# Truthful Aggregation of LLMs with an Application to Online Advertising

**Ermis Soumalias**
University of Zurich
ETH AI Center
ermis@ifi.uzh.ch

**Michael J. Curry**
University of Illinois Chicago
mjc@uic.edu

**Sven Seuken**
University of Zurich
ETH AI Center
seuken@ifi.uzh.ch

## Abstract

The next frontier of online advertising is revenue generation from LLM-generated content. We consider a setting where advertisers aim to influence the responses of an LLM, while platforms seek to maximize advertiser value and ensure user satisfaction. The challenge is that advertisers' preferences generally conflict with those of the user, and advertisers may misreport their preferences. To address this, we introduce MOSAIC, an auction mechanism that ensures that truthful reporting is a dominant strategy for advertisers and that aligns the utility of each advertiser with their contribution to social welfare. Importantly, the mechanism operates without LLM fine-tuning or access to model weights and provably converges to the output of the optimally fine-tuned LLM as computational resources increase. Additionally, it can incorporate contextual information about advertisers, which significantly improves social welfare. Via experiments with publicly available LLMs, we show that MOSAIC leads to high advertiser value and platform revenue with low computational costs. While our motivating application is online advertising, our mechanism can be applied in any setting with monetary transfers, making it a general-purpose solution for truthfully aggregating the preferences of self-interested agents over LLM-generated replies.

## 1 Introduction

*Large language models (LLMs)* are becoming ubiquitous: as coding assistants, chat interfaces, and even alternatives to search engines [Bommasani et al., 2022]. To ensure their usefulness, it is essential to closely align LLM outputs with user preferences. However, in general, there may be multiple interested parties who disagree on the desired behavior of *the same* LLM. This leads to the question of *how to generate LLM replies given multiple conflicting preferences.*

The motivating application for our work is online advertising, the main source of revenue for large tech companies such as Google, Meta, and X. For example, in 2023, Meta's advertising revenue of 132 billion USD constituted more than 97% of its total revenue [Meta, 2024]. Ad auctions are the workhorse mechanism to determine the placement and prices of commercial content [Varian, 2007, Edelman et al., 2007]. LLM providers have begun integrating instant checkout capabilities into their services [OpenAI, 2025], and they are also exploring how to incorporate ads into LLM-generated content [Criddle, 2024]. Thus, new auction mechanisms are needed for this emerging setting.

We present a new auction mechanism for this problem, which we call *MOSAIC (**M**echanism for **O**ptimally **S**ampling and **A**ggregating LLM Outputs with **I**ncentive **C**ompatibility)*. The advertisers are paying, not for some particular item or bundle, but rather to influence the output generated by an LLM in a direction closer to their own preferences. Although MOSAIC could be of interest whenever one has to aggregate the preferences of multiple self-interested agents over LLM behavior (as long as

39th Conference on Neural Information Processing Systems (NeurIPS 2025).

it is reasonable to charge payments), we see online advertising as the most natural setting of interest. For this reason, we refer to participants in the mechanism as *advertisers*.

## 1.1 Problem setting

We consider a situation where a user queries an LLM with a specific question. We assume the following: (i) There is a *reference LLM* that can produce useful replies to the user's query, and (ii) there are *advertisers* who want to influence the reply to the user. In our model, each advertiser is represented via her own LLM or directly with a *reward function*, similar to the function used to fine-tune LLMs in the *Reinforcement Learning from Human Feedback* pipeline [Azar et al., 2023]. For this reason, we refer to an advertiser's value for a reply as her *reward*. The auctioneer's task is, given the user's query and the advertisers' preferences, to *produce a reply that is useful for the user while also generating high rewards for the advertisers*.

## 1.2 Overview of Contributions

In this paper, we present MOSAIC, an auction mechanism designed to aggregate the preferences of multiple self-interested advertisers over LLM-generated replies. The first challenge we address is that MOSAIC must produce replies for which the advertisers receive high rewards, but without steering the LLM's behavior too far from that of the user-centric reference LLM. We address this by drawing a strong connection to the RLHF objective (e.g., Ziegler et al. [2020]), equipping MOSAIC with a hyperparameter that enables the auctioneer to balance between the usefulness of the produced reply to the user and the advertisers in an interpretable and principled way (Section 3.1).

Second, leveraging an importance sampling-based technique, MOSAIC is the only mechanism in the LLM mechanism design literature that both provably converges to the optimal distribution (Corollary 4.1) and converges efficiently in practice (Section 6). Third, MOSAIC can incorporate contextual information, similar to how sponsored search auctions utilize advertiser descriptions, accelerating convergence and increasing value for advertisers and revenue for the auctioneer (Section 6.2).

Fourth, we address technical feasibility and practicality. We adopt the desiderata proposed by Dütting et al. [2024], who argued that auctions must be compatible with existing LLM technology, only using "information obtainable from current models" in a way that is "easy to integrate into the system" and relying only on "easy manipulations of LLM output." Furthermore, it must be computationally feasible to run the auction repeatedly and with different advertiser preferences. In particular, we cannot afford the expensive process of fine-tuning an LLM's weights for each possible query. To address this, we design MOSAIC to work by only post-processing multiple LLM outputs: it requires only "API access", not even viewing the model weights (Section 4.2).

Fifth, we address the fact that advertisers might have an interest in misreporting their preferences (which include their *LLMs*), analogous to over-/under-bidding in traditional auctions. To address this challenge, we employ an allocation rule compatible with Rochet payments [Rochet, 1987]. These uniquely ensure that MOSAIC is strategyproof, i.e., that truthful reporting is a dominant strategy for each advertiser, no matter what the others do (Theorem 5.1). Crucially, Rochet payments ensure that MOSAIC remains strategyproof even if the mechanism has not fully converged to the optimal LLM.

Sixth, we introduce an offset to the Rochet payments to ensure that each advertiser's utility is proportional to her contribution to social welfare. This alignment is crucial for the long-term success of the market, as it incentivizes the participation of only the most relevant advertisers.

Finally, we discuss *individual rationality (IR)*, which guarantees that a participant is weakly better off from participating in the mechanism. In Section 5.2, we discuss the unique properties of our setting that make *ex-post* IR unattainable. However, in Section 5.3.1, we show that MOSAIC is "almost ex-post IR" in a well-defined sense. In Section 6.2, we experimentally show that, MOSAIC is *ex-ante* IR (i.e., advertisers are in expectation better off by participating).

In Section 6, we provide experimental results for the online advertising domain. We demonstrate that MOSAIC quickly converges to the optimal LLM with low computational cost, generating significant value for the advertisers and revenue for the auctioneer while also being useful to the user.

## 2 Related Work

In their pioneering work, Dütting et al. [2024] introduced the field of mechanism design for LLMs, proposing a sequential mechanism where the output sequence is generated token by token, and advertisers bid for their LLM to produce the next token. This work laid the foundation for future work in this area. However, several challenges hinder the adoption of their approach in the real world: (i) Advertisers face the *exposure problem*: small changes in the generated sequence, such as the addition of a word like "not," can completely alter its meaning. An advertiser might pay a significant amount for the tokens generated up to a certain point, only for the continuation to unexpectedly negate or distort her intended message.[1] (ii) The mechanism is easily manipulable if the assumption that advertisers cannot misreport their LLMs is dropped. (iii) For a given prompt, an advertiser's spend grows with the length of the generated sequence. MOSAIC addresses all of the above challenges.

Dubey et al. [2024] proposed a generalization of the position auction [Varian, 2007, Edelman et al., 2007] to a setting where each advertiser is interested in having some specific text ad displayed alongside the organic results. In their mechanism, an LLM module coupled with an auction module work in tandem to merge the ads into a single ad summary in an incentive-compatible way.

Hajiaghayi et al. [2024] considered advertisers bidding in an auction to have their ads placed into various portions of a retrieval-augmented generation (RAG) response. Their auction only allows advertisers to report (and misreport) scalar values to influence the inclusion of non-misreportable advertising texts in the combined output. This is in contrast to MOSAIC, which allows the aggregation of full LLMs that advertisers may arbitrarily misreport.

In work that appeared after the initial version of this paper, Bergemann et al. [2024] study a mechanism design problem where agents have both private types and signals, one motivation of which is to model a generalization of our problem. For further discussion of related work, see Appendix A.

## 3 Framing Sequence Generation as a Mechanism Design Problem

### 3.1 Formal Model

A *user* issues a query $x$. There is a *reference LLM* $\pi_{\text{ref}}$ that the auctioneer aims not to deviate from too much (e.g., because it is responsible for providing useful replies to the user). Additionally, there is a set $N$ of $n$ advertisers who have their own preferences for the reply (i.e., a token sequence) that will be returned to the user. We use the terms *sequence* and *reply* interchangeably.

We let $r_i(x, y)$ denote advertiser $i$'s *reward* for reply $y$, given query $x$.[2] Informally, the auctioneer's goal is to sample the final reply from a distribution that optimizes the advertisers' expected rewards without substantially diverging from $\pi_{\text{ref}}$. This is motivated by traditional online ad auctions, where the implicit goal is to maximize advertiser value subject to ensuring a chosen level of usefulness to the user (e.g., the balance between allocating search slots to ads and native content). In the context of an LLM environment, the analogous notion of "usefulness to the user" is represented by the closeness of the final distribution to $\pi_{\text{ref}}$. Formally, the goal is to choose $\pi$ to maximize:

$$\mathbb{E}_{y \sim \pi}\left[\sum_{i \in N} r_i(x, y)\right] - \tau D_{\text{KL}}(\pi(\cdot|x)||\pi_{\text{ref}}(\cdot|x)), \tag{1}$$

where $\tau > 0$ is a hyperparameter enabling the auctioneer to balance producing replies closer to the reference LLM or with higher reward for the advertisers, and $D_{\text{KL}}$ is the Kullback-Leibler divergence.

This objective is analogous to the standard Reinforcement Learning from Human Feedback (RLHF) approach [Ziegler et al., 2020], but replaces the human feedback reward function with the advertisers' aggregate reward $r(x, y) = \sum_{i \in N} r_i(x, y)$. For an overview of RLHF, see Rafailov et al. [2023, §3].

---

[1]As an example, suppose advertiser A wins the bids for all the tokens in the sequence "Planning your next vacation? For the cheapest flights book via..." However, she loses the bid for the crucial next token, her brand name. She thus pays for the preceding tokens without benefiting from associating her brand with the message.

[2]In theory, an advertiser's reward for a generated reply could also depend on user-specific information, but we abstract that away. Equivalently, we assume that the advertisers provide user-specific reward functions.

The optimal solution $\pi_r^*$ to the optimization problem in (1) was derived by Peters and Schaal [2007]:

$$\pi_r^*(y|x) = \frac{1}{Z(x)} \pi_{\text{ref}}(y|x) \exp\left(\frac{1}{\tau} \sum_{i \in N} r_i(x, y)\right), \tag{2}$$

where $Z(x) = \sum_{y \in T^*} \pi_{\text{ref}}(y|x) \exp\left(\frac{1}{\tau} \sum_{i \in N} r_i(x, y)\right)$ is the partition function.

The performance goal we seek to maximize is $\tau \log \pi_{\text{ref}}(y|x) + \sum_{i \in N} r_i(x, y)$; as it aggregates the interests of both the user and all the advertisers, we refer to it as the *social welfare* of $y$.

Let $\hat{R}$ be the set of all possible reports by the advertisers. A *mechanism* is defined as a pair $(\pi, p)$. The *allocation rule* $\pi : \hat{R} \to (T^* \to \Delta(T^*))$ maps any report profile $\hat{r} = (\hat{r}_1, \hat{r}_2, \dots, \hat{r}_n) \in \hat{R}$ of the advertisers' rewards to an LLM $\pi_{\hat{r}}$, which in turn is a mapping from a user query $x$ to a distribution over token sequences $\delta(T^*)$. We denote the LLM that the allocation rule $\pi$ induces for reports $\hat{r}$ as $\pi_{\hat{r}}$, and the optimal LLM for those reports (i.e., the maximizer of Equation (1)) as $\pi_{\hat{r}}^*$. The *payment rule* $p : \hat{R} \to \mathbb{R}^n$ maps any report profile $\hat{r}$ to a payment profile $p(\hat{r})$, where $p_i(\hat{r})$ is the payment of the $i$-th advertiser. We aim for *strategyproof* mechanisms, meaning that no advertiser has an incentive to misreport her preferences:

**Definition 3.1** (Strategyproof). *A mechanism $(\pi, p)$ is* strategyproof *if, for any advertiser $i \in N$, true reward function $r_i$, reported reward function $\hat{r}_i$, reported reward functions $\hat{r}_{-i}$ by the other advertisers and prompt $x$: $\mathbb{E}_{y \sim \pi_{(r_i, \hat{r}_{-i})}(\cdot|x)}[r_i(x, y) - p_i(r_i, \hat{r}_{-i})] \geq \mathbb{E}_{y \sim \pi_{(\hat{r}_i, \hat{r}_{-i})}(\cdot|x)}[r_i(x, y) - p_i(\hat{r}_i, \hat{r}_{-i})]$, where $r_i(x, y) - p_i(\hat{r}_i, \hat{r}_{-i})$ is the utility of advertiser $i$ for reply $y$ when her payment is $p_i(\hat{r}_i, \hat{r}_{-i})$.*

### 3.2 The shortcomings of VCG in this setting

At first sight, it may seem that the Vickrey–Clarke–Groves (VCG) mechanism would be suitable for our setting [Vickrey, 1961, Clarke, 1971, Groves, 1973]. VCG selects the outcome that maximizes the sum of all agents' values. This can be either a single optimal sequence for Equation (1) or the optimal distribution of Equation (2). The VCG mechanism has a corresponding payment rule to incentivize truthful reporting: it charges each agent her externality, the total reduction in value (respectively expected value) for the other agents that her participation in the mechanism caused.

However, in our setting, VCG is *not* a viable option: VCG requires calculating the *exact* optimal solution to the optimization problem, which is intractable for choosing an LLM to maximize Equation (1) and is even difficult for choosing a single optimal sequence. If a sub-optimal solution is chosen, VCG's strategyproofness is no longer guaranteed [Nisan and Ronen, 2007, 1999, Lehmann et al., 2002]. We provide an intuitive example illustrating this failure for two advertisers in Appendix B.1.

## 4 The MOSAIC Mechanism: Allocation Rule

### 4.1 Convergence to Optimality, Advertiser Contexts and Importance Sampling

In this section, we introduce MOSAIC's allocation rule. The high-level idea is as follows: first, a set of $M$ *candidate replies* is generated based on an LLM $\pi_{\text{gen}}$. Then, the probability of returning each candidate is re-weighted based on the advertisers' reports and the reference LLM $\pi_{\text{ref}}$, so that as $M \to \infty$, the return probability of each reply converges to that under the optimal distribution of Equation (2). This approach resembles importance sampling techniques that have been used in various LLM training pipelines (e.g., Xie et al. [2023]). All proofs are deferred to Appendix B.

**Corollary 4.1.** *For any reported reward functions $r \in R$ and any LLM $\pi_{gen}$ such that $\pi_{ref}$ is absolutely continuous with respect to $\pi_{gen}$, the MOSAIC policy $\pi_{r,M}(\cdot|x)$ from Algorithm 1, using $M$ candidate replies, converges to an optimal solution for the platform's objective (Equation (1)) as $M \to \infty$. Formally, $\lim_{M \to \infty} \pi_{r,M}(\cdot|x) = \pi_r^*(\cdot|x) \in \arg\max_{\pi \in \Delta(T^*)} \mathbb{E}_{y \sim \pi(\cdot|x)}[r(x, y)] - \tau D_{KL}(\pi || \pi_{ref})$.*

Based on Corollary 4.1, MOSAIC converges to the optimal distribution for *any* LLM $\pi_{\text{gen}}$, provided that $\pi_{\text{ref}}$ is absolutely continuous with respect to $\pi_{\text{gen}}$. While $\pi_{\text{gen}} = \pi_{\text{ref}}$ is an intuitive choice, it would result in impractically slow convergence rates. The reason is that advertisers have high rewards for responses that explicitly mention their brands, but $\pi_{\text{ref}}$ considers replies with mentions of specific brands extremely unlikely. Consequently, generating candidate replies directly from $\pi_{\text{ref}}$ leads Algorithm 1 to sample from a set of low-reward candidates, hindering performance.

---

**Algorithm 1:** Allocation Rule for MOSAIC

---

**Input:** User prompt $x$, reference LLM $\pi_{\text{ref}}$, LLM used for candidate reply generation $\pi_{\text{gen}}$, advertiser reward functions $\{r_i\}_{i=1}^n$, number of candidate replies to generate $M$, reference LLM weight $\tau$

**Output:** Reply $y$ drawn according to the optimal distribution as defined in Equation (1) for the aggregate reward function $r(x, y) = \sum_{i=1}^N r_i(x, y)$

**1** Sample $y_j \sim \pi_{\text{gen}}(\cdot|x),\ 1 \le j \le M$

**2** Calculate $r(x, y_j) = \sum_{i=1}^N r_i(x, y_j),\ 1 \le j \le M$

**3** **return** $y \sim \text{softmax}\left(\frac{r(x,y_1)}{\tau} + \log\frac{\pi_{\text{ref}}(y_1|x)}{\pi_{\text{gen}}(y_1|x)}, \ldots, \frac{r(x,y_M)}{\tau} + \log\frac{\pi_{\text{ref}}(y_M|x)}{\pi_{\text{gen}}(y_M|x)}\right)$

---

To address this challenge, we generate candidate sequences not from $\pi_{\text{ref}}(\cdot|x)$, but instead from a *context-aware LLM*, $\pi_{\text{con}}(\cdot|x; c)$. The instance-specific context $c$ is designed to bridge the gap between the parts of the output space favored by $\pi_{\text{ref}}$ and those valued by the advertisers.

We formalize this intuition in Appendix B.3, connecting MOSAIC's allocation rule to an importance-based sampling estimator for $\pi_r^*$ and proving the following lemma:

**Lemma 4.2.** *For any LLM $\pi_{gen}$ such that $\pi_{ref}$ is absolutely continuous with respect to $\pi_{gen}$, the variance of $\pi_{r,M}(\cdot|x)$ as an estimator of $\mathbb{E}_{y \sim \pi_r^*(\cdot|x)}[\pi_r^*(y|x)]$ is*

$$\frac{1}{M}\left(\sum_{y \in Y} \frac{\pi_r^*(y|x)^4}{\pi_{gen}(y|x)} - \left(\sum_{y \in Y} \pi_r^*(y|x)^2\right)^2\right).$$

Applying Chebyshev's inequality to the variance of Lemma 4.2, MOSAIC converges to $\pi_r^*$ at a rate of $\sqrt{M}$ (Lemma B.3).[3] However, this convergence rate also depends on $\frac{\pi_r^*(y|x)^4}{\pi_{\text{gen}}(y|x)}$. Thus, generating replies via an LLM closer to $\pi_r^*$ reduces the estimator's variance and improves convergence speed.

In our application of integrating advertisers' interests into LLM outputs, $c_i$ is a context-specific description of the $i$-th advertiser. These descriptions, supplied by the advertisers themselves, should be easily verifiable and factually accurate, akin to "MusicMastery: offering online music lessons", or "InstaTune: selling musical instruments."[4] This approach is analogous to search engine optimization in sponsored search advertising, where advertisers supply and potentially optimize their own descriptions to influence how they are presented by the auction mechanism.

Our experiments in Section 6.2 demonstrate that using the context-aware LLM to generate candidate replies achieves substantially higher rewards and utility for the advertisers, increased revenue for the auctioneer, and faster convergence. In the rest of the paper, we refer to using the reference and context-aware LLMs as the baseline and context-aware versions of our mechanism, respectively.

### 4.2 Practical Considerations

**Input Methods and Black-Box Access to Advertiser LLMs.** MOSAIC's allocation and payment rules do not depend on the advertisers' full reward functions, but only on their rewards for the candidate replies. Thus, MOSAIC requires only "API access" to the involved LLMs without fine-tuning or access to their weights. Rafailov et al. [2023] established a mapping between an agent's LLM and her implicit reward function, allowing MOSAIC to use as inputs reply probabilities (i.e., LLM inference calls) instead of rewards. For more details, see Appendix C.2.

**Static Setting and Incentive Implications.** The number $M$ of candidate replies considered by MOSAIC is predetermined. Each advertiser observes all $M$ candidate replies before submitting her reports, resulting in a *static* rather than a *dynamic* setting (e.g., as in Dütting et al. [2024]), where advertisers interact sequentially with evolving information, significantly complicating incentive considerations. Theorem 5.1 ensures that truthful reporting is a dominant strategy for each advertiser.

---

[3]In Appendix B.4 we establish a more general result, showing that for any sequence $y$, its estimated probability converges to the corresponding probability under $\pi_r^*$ at a rate of $1/\sqrt{M}$.

[4]A practical way of implementing $\pi_{\text{con}}(\cdot|x; c)$ given $\pi_{\text{ref}}(\cdot|x)$ is to augment the input $x$ to the reference LLM with the advertiser descriptions. In our example: "Try to mention ⟨advertiser x⟩, ⟨advertiser x description⟩.'

**Complexity.** Generating a single candidate reply of length $L$ tokens requires $L$ forward passes through $\pi_{\text{gen}}$, as tokens are generated sequentially in an autoregressive manner. Evaluating each reply processes the entire sequence in a single forward pass (and is compatible with efficient evaluation methods, e.g. Li et al. [2024]). For $M$ candidate replies and $n$ advertisers, MOSAIC requires $M \cdot (L + n + 1)$ total forward passes. In contrast, Dütting et al. [2024] require $L \cdot n$ forward passes, as all $n$ advertiser LLMs participate in generating each token. Section 6 shows that MOSAIC achieves convergence with $M = 20$ candidate replies, reducing computational cost for large $n$.

**Compatibility with Efficient Methods and Practical Costs.** MOSAIC relies solely on LLM forward passes, the core operation modern architectures are optimized for, enabling it to take full advantage of existing optimizations for efficient generation. As a result, MOSAIC converges to the optimal distribution with compute costs equivalent to just five LLM queries (Appendix D.8).

**Parallelization and User-Perceived Latency.** The generation and evaluation of each candidate sequence are independent processes, allowing MOSAIC to be fully parallelized. In a fully parallelized setting, the response time for a user query is comparable to directly querying a *single* LLM.

**Inherent Competition.** Unlike conventional auctions, MOSAIC ensures high baseline revenue even in low-competition environments, as advertisers always compete against $\pi_{\text{ref}}$. Traditional auctions rely on reserve prices to boost revenue, but a slight miscalibration above a critical threshold can result in zero revenue. In contrast, MOSAIC uses a *single* tunable parameter, $\tau$, and its revenue varies smoothly and differentiably with $\tau$ (see the proof of Theorem 5.1). As a result, revenue optimization in MOSAIC is significantly more robust to parameter tuning.

**An "Autobidder" Perspective on the Mechanism.** In practice, an advertiser might not have the know-how to produce an LLM reflecting their preferences. In conventional online ad auctions, two existing practices are used to solve a similar problem: platforms offer tools to help advertisers improve ad quality, and they provide autobidders to help advertisers bid. The largest online advertising platforms have the capacity to host and finetune LLMs, so they already have the infrastructure to offer an "LLM autobidder." Concretely, they could train an LLM for an advertiser, reflecting the advertiser's interests, and they could also provide inference. Even under this "autobidder" implementation, all properties of our mechanism are preserved.

## 5   The MOSAIC Mechanism: Payment Rule

In this section, we first show how the allocation rule from Section 4 can be combined with an appropriate payment rule so that the resulting mechanism is strategyproof (Section 5.1). Then, we detail how auctions for LLM-generated content differ from standard auctions (Section 5.2). Taking those differences into account, we create a payment offset, so that MOSAIC is both strategyproof and social welfare-aligned (Section 5.3). We defer all proofs to Appendix C.

### 5.1   Strategyproofness through Cyclic Monotonicity

The allocation rule of Algorithm 1 satisfies *cyclic monotonicity* [Rockafellar, 1970, § 24]. Rochet [1987] first proposed the use of cyclic monotonicity in mechanism design as a generalization of monotonicity in single-parameter settings [Myerson, 1981]. For general settings, cyclic monotonicity of the allocation rule is a sufficient and necessary condition for us to prove:

**Theorem 5.1.** *The allocation rule of Algorithm 1 can be combined with a payment rule such that for any advertiser $i \in N$ and set of candidate replies $\{y_j\}_{j=1}^M$, reporting truthfully is a dominant strategy. Advertiser $i$'s expected utility (up to a constant of integration $C$) under truthful reporting is:*
$$\tilde{U}_i(r_i, \hat{r}_{-i}; \pi_{ref}, \pi_{gen}) = C + \tau \log \left( \sum_{j=1}^M \exp \left( \frac{\sum_{k \in N \setminus \{i\}} \hat{r}_k(x, y_j) + r_i(x, y_j)}{\tau} \right) + \log \frac{\pi_{ref}(y_j | x)}{\pi_{gen}(y_j | x)} \right).$$

Note that, based on Theorem 5.1, in MOSAIC it is *always* a dominant strategy for an advertiser to report truthfully. Crucially, this is not the case for VCG, where truthful reporting would be optimal only if the allocation rule had converged to the optimal distribution (Example 1).

## 5.2 Differences from Standard Auction Settings

Auction mechanisms designed to sell items or ad slots typically rely on simplifying assumptions that do not apply in a setting with LLM-generated content. Key differences include: (i) *Non-Negative Values:* These mechanisms assume agents' values are non-negative due to having zero value for the empty bundle and free disposal. In contrast, in our setting, an advertiser's reward can be negative based on the discrepancy between her LLM and the reference LLM. (ii) *Advertiser-Specific Allocations:* These mechanisms allocate different item bundles to different agents. Here, a single reply is produced (iii) *Zero Utility for Non-Participation:* In most auction settings, not participating yields zero utility. Here, non-participation can result in negative utility since the produced reply may be unfavorable to non-participating advertisers.[5] For details, see Appendix C.2.

## 5.3 Advertiser-specific Utility Offset

We now modify MOSAIC's payment rule by adding a *payment offset* (and thus a utility offset): $C = -\tilde{U}_i(0, \hat{r}_{-i}; \pi_{ref}, \pi_{con})$. Informally, we additionally charge each advertiser her utility in Theorem 5.1 if her reward for all candidate replies was zero. This offset maintains the key properties of our mechanism (i.e., strategyproofness and convergence to the optimal distribution), while also achieving two additional properties that are critical for the long-term success of a market for LLM aggregation. The first is *"Almost IR:"* An advertiser with weakly positive reward for all candidate replies has weakly positive expected utility for all reports by the other advertisers. The second is *"What you give is what you get:"* an advertiser's expected utility is monotone in how well-aligned her exponentiated reward for the replies is with the interim allocation rule if she were to not participate.

### 5.3.1 "Almost Individually Rational"

Individual rationality is important to incentivize agents to participate in the mechanism. In Appendix C.3, we explain how the unique properties of our setting, namely lack of free disposal and a common outcome for all agents, make the standard notion of individual rationality (i.e., weakly positive utility from participation) unobtainable while converging to the optimal distribution and maintaining strategyproofness. Then, we explain how, our payment offset achieves "almost IR:" In Lemma C.1 we prove that the expected utility of an advertiser who has zero reward for all candidate replies and bids truthfully is zero, i.e., advertisers that do not contribute to the social welfare (but also do not detract from it) have zero utility. Similarly, in Lemma C.2 we prove that if an advertiser's reward for all candidate sequences is (weakly) positive, then her expected utility is (weakly) positive.

**Remark 1.** *In Section 6, we experimentally show that our offset payment rule, coupled with our context-aware allocation rule, results in both high expected rewards and positive utility for the advertisers (i.e., ex-ante individual rationality), as well as significant revenue for the auctioneer.*

### 5.3.2 "What you give is what you get"

Our allocation rule, which is the only one over a finite set of replies that converges to the optimal LLM, is also the (sub)gradient of the utility to ensure truthfulness [Rochet, 1987]. Because the allocation rule is the same for all advertisers, their utilities must also be the same, up to advertiser-specific offsets, as indicated by Theorem 5.1. However, not all advertisers contribute equally to social welfare. In Appendix C.4, we explain why implementing the mechanism without carefully designed offsets would lead to free-riding. Specifically, if an advertiser's expected utility is positive, even when her relevance to the user query is low, it incentivizes her participation. This dilutes the utility for all participants, discouraging more relevant advertisers from participating. As a result, the expected rewards for advertisers decline, and the generated replies become less useful for users. To avoid free-riding, it is crucial to align each advertiser's utility with her contribution to social welfare.

**Lemma 5.2.** *For the offset $C = -\tilde{U}_i(0, \hat{r}_{-i}; \pi_{ref}, \pi_{con})$ advertiser $i$'s expected utility under truthful reporting is $U_i(r_i, \hat{r}_{-i}; \pi_{ref}, \pi_{con}) = \tau \log \left( \sum_{j=1}^{M} \exp \left( \frac{r_i(x, y_j)}{\tau} \right) \pi_{\hat{r}_{-i}}(y_j \mid x) \right).$*

---

[5]While we focus on the most general model that includes negative rewards, MOSAIC maintains all its desirable properties discussed in Sections 4 and 5 under the assumption that all advertiser rewards are nonnegative.

| User query |
| --- |
| What are effective ways to learn a musical instrument online? |

| Advertisers |
| --- |
| - MusicMastery: offering online music lessons and virtual tutoring
- InstaTune: selling musical instruments and learning aids |

| Context-Aware Mechanism Output |
| --- |
| Effective ways to learn a musical instrument online include:
1. Online music lessons: Platforms like **MusicMastery** offer one-on-one virtual tutoring, group classes, and pre-recorded lessons. These lessons can be tailored to your skill level and learning style.
2. Interactive learning tools: **InstaTune** offers a range of interactive learning tools, such as apps, software, and games, that make learning fun and engaging... . |

| Baseline Mechanism Output |
| --- |
| - Practice consistently and regularly: Consistent practice is essential to learning any musical instrument, and it's even more important when learning online. Set aside a specific time each day or week to practice, and make sure to stick to it.
- Find a good teacher or mentor: While it's possible to learn some things on your own, working with a good teacher or mentor can help you improve faster and avoid developing bad habits. . . . |

Figure 1: Comparison of context-aware and baseline mechanism outputs using 10 candidate replies.

In words, Lemma 5.2 shows that each advertiser's expected utility is monotone in how well aligned her exponentiated reward function is with MOSAIC's induced LLM $\pi_{\hat{r}_{-i}}$ (i.e., the optimal LLM for the reports $\hat{r}_{-i}$ that would have been obtained had advertiser $i$ not participated).

**Remark 2.** *In Section 6.2, we experimentally show that the offset of Section 5.3 induces a strong positive correlation between an advertiser's contribution to social welfare and her expected utility gain from participating. Moreover, for the tested distribution of instances, MOSAIC satisfies ex-ante IR.*

# 6 Experiments

In this section, we evaluate MOSAIC's performance on its flagship application of online advertising.

## 6.1 Experiment Setup

We create synthetic instances, each comprising a user query (e.g., "How to learn a musical instrument online?") and two advertisers (e.g., "MusicMastery, offering online music lessons"). This matches the setup of Dütting et al. [2024] while highlighting MOSAIC'S performance and revenue, even in low competition scenarios. We use Llama-2-7b-chat-hf [Touvron et al., 2023] as the base architecture for all LLMs. In Appendices D.6 and D.10 we extend our analysis to settings with more advertisers and alternative architectures, observing similarly strong results. See Appendix D for details.

## 6.2 Experimental Results

In this section, we evaluate MOSAIC's overall effectiveness, assessing both its allocation and payment rules. We first focus on how well the allocation rule generates value for advertisers and converges to the optimal distribution, comparing the baseline and context-aware versions. Figure 1 illustrates how the context-aware mechanism enhances response relevance for advertisers. Notably, only the context-aware mechanism successfully incorporates advertisers into the replies.

In Figure 2a, we plot the log probability of the replies returned by MOSAIC with respect to the optimal distribution (Equation (2)) against the number $M$ of candidate replies generated. We compare the context-aware version of MOSAIC to the baseline version to evaluate the effectiveness of incorporating contextual information. Note that baseline MOSAIC is the *strongest applicable benchmark*, as it is the *only* other tractable mechanism for this problem in the literature that converges to the optimal distribution (for a detailed discussion, please see Appendix D.2).

To assess MOSAIC's convergence, we estimate the log probability of sampling a *single reply* from the optimal distribution.[6] We observe that for both the context-aware and baseline versions of

---

[6]The closed-form solution of Equation (2) allows us to evaluate the probability of sentences with respect to the optimal solution, but it does not enable us to sample from that distribution, which would require using RL to train the optimal LLM on the advertisers' aggregate reward function, which is computationally infeasible for

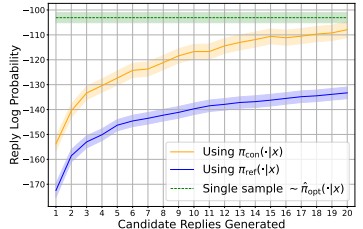 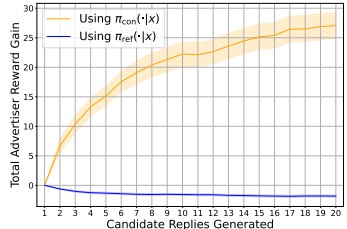 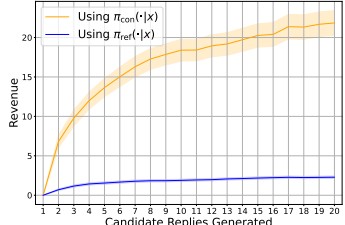

(a) Returned reply log probability as a function of the number of candidate replies generated using $\pi_{\text{ref}}$ and $\pi_{\text{con}}$. We also show a proxy of drawing a single sequence from the optimal distribution.

(b) Total advertiser normalized reward as a function of the number of candidate replies generated using $\pi_{\text{ref}}$ and $\pi_{\text{con}}$.

(c) Revenue as a function of the number of replies generated using $\pi_{\text{ref}}$ and $\pi_{\text{con}}$.

Figure 2: Reply log probability, total advertiser normalized reward, and revenue as a function of the number of candidate replies generated using $\pi_{\text{ref}}$ and $\pi_{\text{con}}$. Averages over 1250 runs with 95% CIs.

MOSAIC, the log probability of the returned reply scales rapidly with the number of candidate replies. This aligns with our analysis in Section 4, where we proved that both versions converge to the optimal distribution. We also observe that incorporating context significantly accelerates convergence. Notably, the context-aware version of MOSAIC achieves higher log probability with respect to the optimal distribution with four candidate replies than the baseline version with 20. Additionally, with only 20 candidate replies, the context-aware mechanism can almost match our estimate of the log probability of sampling from the optimal distribution. In Appendix D.4, we further demonstrate that the usefulness to the user also scales rapidly with the number of candidate replies.

In Figure 2b, we plot the total expected reward for advertisers against the number of candidate replies generated, with the expectation over the distribution of the returned reply from the set of candidate replies. To make the plot more interpretable, we normalize each advertiser's expected reward by her expected reward had she not participated. Specifically, we plot $r_{i,\text{norm}}(x) = \sum_{j \in M} r_i(x, y_j) \cdot \pi_r(y_j | x) - \sum_{j \in M} r_i(x, y_{-i,j}) \cdot \pi_{r_{-i}}(y_{-i,j} | x)$, where $y_{-i}$ are the candidate replies that would have been generated if advertiser $i$ had not participated, and $\pi_{r_{-i}}$ MOSAIC's allocation rule in that case.[7] We compare MOSAIC's two versions, highlighting the value of incorporating context.

Figure 2b demonstrates that our context-aware mechanism significantly boosts advertiser rewards, with benefits scaling rapidly with the number of candidate replies. In contrast, the baseline version fails to improve advertiser rewards and may even cause a slight decrease: generating more replies adds marginal positive reward to the reference LLM (see Appendix D.4), but does so at the expense of advertisers. In summary, MOSAIC's support for context-aware LLMs allows it to quickly converge to the optimal distribution and generate substantial rewards for advertisers. In an ablation study (Appendix D.7), we demonstrate that MOSAIC maintains strong performance across all key metrics even under significantly reduced contextual input.

In Appendix D.8 we provide a detailed experimental evaluation of MOSAIC's compute requirements, showing how it requires the compute cost of just five LLM queries to converge, and discuss how it can be fully parallelized so that the user-perceived latency is the same as a *single* LLM query.

Next, we examine the impact of our payment rule from Section 5. In Figure 2c, we plot the auctioneer's revenue for both the context-aware and baseline versions of MOSAIC. Our payment rule in combination with the context-aware allocation rule yields revenue that scales rapidly with the number of candidate replies. Moreover, comparing the advertisers' total payment with their normalized reward in Figure 2b, we see that for the context-aware mechanism, our payment rule converts much of the advertisers' surplus into revenue, while also ensuring positive advertiser utility.

These results align with our theoretical analysis. In Section 5.3.1, we showed that an advertiser with positive rewards for candidate replies—as in the context-aware mechanism—achieves positive expected utility, and in Section 5.3.2 we introduced the "what you give is what you get" property. In

the number of problem instances we test. Instead, we generate replies from $\pi_{\text{ref}}$ and evaluate them based on the induced probabilities of $\pi_{\text{ref}}$, for which $\pi_{\text{ref}}$ is the optimal LLM. This serves as a proxy for the log probabilities we should expect if we were to draw replies from the optimally fine-tuned model for each query.

[7] To reduce computational costs, we estimate an advertiser's reward for not participating based on her expected reward over the already generated replies in which her brand is not mentioned by name, motivated by the fact that if she does not participate, her brand will not be mentioned.

Appendix D.5 we show that, for both versions of our mechanism, the payment offset increases the advertisers' expected utility, and makes the relationship between an advertiser's contribution to social welfare and utility significantly more linear and positively correlated. Concretely, our offset increases the Pearson correlation between advertiser utility and reward from $0.4$ to $0.8$.

## 7 Discussion

The motivation for our paper is one particularly common way in which users interact with LLMs: using them as search tools with a single query. This use of LLMs is a close cousin to ordinary search, and most of the existing considerations for sponsored search also apply. For example, the platform would need to vet participating advertisers as well as the content they want to generate, and the platform would also need to visually distinguish advertiser-influenced parts of the response.

We have made several design choices in our problem setting. While these are completely standard in mechanism design, and we think they also make sense in the application we envision, they still represent a limitation in the scope of our work. Below, we discuss those design choices in more depth, and take the opportunity to consider broader impacts and possible directions for future work.

The first such design choice is the platform's objective in Equation (1). This objective corresponds to a "utilitarian" concept of social welfare. This is the most common objective in mechanism design, and indeed, all large online ad platforms maximize for this objective. However, a platform might reasonably focus on other objectives, for example, revenue or egalitarian fairness. As long as a closed-form solution for the platform's objective can be derived and the corresponding allocation rule satisfies cyclic monotonicity, our mechanism is also applicable for that objective.

A second design choice is our focus on a single-query, static mechanism design setting. Handling multi-query LLM interactions involving lengthy conversations would include a much larger range of usage patterns. Determining where and how advertising could be appropriate (e.g., in product searches) and where it might be inappropriate (e.g., in emotionally challenging conversations) would require careful consideration. Additionally, this would involve dynamic mechanism design. In Section 2, we explain some limitations of the specific dynamic mechanism proposed by Dütting et al. [2024] for this problem. However, to a large extent, these issues are inherent to any dynamic mechanism and not unique to that paper; overcoming them is an important direction for future work.

Finally, we have made the design choice of equating the usefulness of a reply to the user with the likelihood of that reply given by the reference LLM, and we have equated an advertiser's value for a reply with the induced reward of that reply by their LLM. Regarding the first choice, we argue that it is reasonable to assume that the platform wants its results to be as high-quality as possible. The second choice corresponds to the standard assumption in mechanism design that participants know their type for any possible outcome. But in practice, it may not be realistic for advertisers to train their LLMs and run inference on potentially private user data. In Section 4.2, we have discussed an autobidder perspective on the mechanism that addresses both of these challenges.

## 8 Conclusion

We have introduced MOSAIC, a novel auction mechanism for aggregating preferences over LLM outputs. MOSAIC provably converges to the theoretically optimal distribution and it also facilitates a principled method for balancing participants' expected rewards with the divergence from a reference LLM. Thus, our mechanism is particularly well-suited for online advertising, allowing the integration of advertiser LLMs with a reference LLM responsible for generating user-centric replies.

Our payment rule removes any incentive to misreport preferences, achieving the central mechanism design goal of strategyproofness. While ex-post IR is incompatible with strategyproofness in our setting, we experimentally show that our mechanism is ex-ante individually rational and "almost individually rational" in a certain sense. Furthermore, it ensures that each participant's utility gain is proportionate to her contribution to social welfare, an essential alignment property in this setting.

Experimentally, we have demonstrated that by incorporating contextual information, MOSAIC's outputs rapidly converge to the optimal distribution, generating significant value for the advertisers while also effectively recapturing a considerable portion of this value as revenue. These findings demonstrate the practical efficacy and potential of our approach in realistic settings.

## Acknowledgments

We would like to thank various anonymous reviewers for their thoughtful comments and suggestions. We are also thankful for the feedback we received from participants at the following events: the 2025 NBER Market Design Working Group Fall Meeting, the Foundation Models and Game Theory workshop at EC'24, the Harvard EconCS seminar, the UChicago CS department seminar, the Frontiers of Online Advertising Workshop at EC'24, and the 2024 Rising Stars in Market Design workshop. This paper is part of a project that has received funding from the European Research Council (ERC) under the European Union's Horizon 2020 research and innovation program (Grant agreement No. 805542).

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

# A   Further related work

[Feizi et al. 2024] presented an abstract design for an LLM advertising system and detailed a number of research challenges that would have to be overcome in the course of implementation. [Conitzer et al. 2024] drew connections between *computational social choice* and LLM alignment. [Fish et al. 2023] presented work in the opposite direction: how can LLMs be used to solve problems in social choice? [Harris et al. 2024] studied Bayesian persuasion in an abstract setting where a "simulator" (for example, a realistic LLM) of the agent is available. [Werner et al. 2024] gave preliminary empirical evidence that LLM-generated content can influence consumers. [Banchio et al. 2025] discuss running auctions where the allocative decision involves placing ads in a multi-turn conversation after up-front bidding.

In work that appeared after the initial version of this paper, [Bergemann et al. 2024] considers a setting where bidders have both private types reflecting their preferences and private signals reflecting information about a world state (for example, user interests). When applied to LLM auctions, they identify their setting as a generalization of ours, where the signals known to the advertisers may actually improve the output for the user, and where deviations from the reference LLM may be measured by arbitrary distance functions. They analyze conditions under which a welfare maximizing mechanism can be implemented truthfully (eliciting both types and signals) in posterior equilibrium, but do not consider practical implementations of their mechanism.

# B   Details from Section 4

In this section, we present all omitted theorems and proofs from Section 4.

## B.1   An Example Where VCG Is Not Strategyproof in This Setting

In Example 1, we present a simple example illustrating that the VCG mechanism is not strategyproof when the allocation rule cannot fully converge to the optimal LLM based on the advertisers' reports.

**Example 1.** *Suppose there are two advertisers, $A$ and $B$, and five possible replies:*

- *two replies that mention both advertisers,*

- *one reply that mentions only $A$,*

- *one reply that mentions only $B$,*

- *and one reply that mentions neither advertiser.*

*The advertisers' rewards for each reply are shown in Table 1:*

| Outcome | $r_A$ | $r_B$ |
|---|---|---|
| $\{A, B\}_1$ | 0.9 | 0.9 |
| $\{A, B\}_2$ | 0.4 | 0.4 |
| $\{A\}$ | 1.0 | 0.0 |
| $\{B\}$ | 0.0 | 0.5 |
| $\emptyset$ | 0.0 | 0.0 |

Table 1: Advertiser rewards for different replies.

*Assume that $\tau = 0$, so that the optimal LLM would deterministically select $\{A, B\}_1$, maximizing total advertiser reward. However, suppose that the allocation rule cannot converge to this reply—for instance, because RLHF is used to train the aggregate LLM $\pi_{\hat{r}}$ and the process is noisy. Instead, when aiming for $\{A, B\}_1$, the resulting LLM $\pi$ produces $\{A, B\}_2$.[8]*

*Under truthful reports, advertiser $A$'s utility would be*

$$u_A = r_A(\{A, B\}_2) + r_B(\{A, B\}_2) - r_B(\{B\}) = 0.4.$$

---

[8]Alternatively, one could interpret this setup as a setting where the mechanism cannot converge to replies mentioning both advertisers, perhaps due to optimization difficulties, but can converge to replies mentioning only one advertiser.

*However, if advertiser A misreports by declaring $\hat{r}_A(\{A,B\}_1) = 0.0$, the mechanism would instead converge to an LLM that always returns $\{A\}$. In that case, A's utility would be*

$$u_A = r_A(\{A\}) + r_B(\{A\}) - r_B(\{B\}) = 0.5.$$

*Thus, advertiser A would be incentivized to misreport her preferences, illustrating how VCG payments cannot ensure strategyproofness when the allocation chosen may be suboptimal.*

## B.2  Proving that MOSAIC converges to the target distribution in the limit

**Theorem B.1.** *Let $\pi_{r,M}(y|x)$ be the probability of sampling output sequence $y$ for input sequence $x$ according to Algorithm 1, where $M$ is the number of candidate sequences generated and $\hat{r} \in \hat{R}$ is the advertisers' reported reward functions. For any LLM $\pi_{gen}$ such that $\pi_{ref}$ is absolutely continuous with respect to $\pi_{gen}$, the policy induced by MOSAIC approaches the following limit:*

$$\lim_{M \to \infty} \pi_{\hat{r},M}(y|x) = \pi_{ref}(y|x) \frac{\exp(\hat{r}(x,y)/\tau)}{\mathbb{E}_{y' \sim \pi_{ref}(\cdot|x)}[\exp(\hat{r}(x,y')/\tau)]} \tag{3}$$

*Theorem B.1 Proof.* Let $\pi_{\hat{r},M}(y|x, \{y_j\}_{j=1}^M)$ be the probability of returning output sequence $y$ for input sequence $x$ according to Algorithm 1 conditioned on the set of generated candidate sequences being $\{y_j\}_{j=1}^M$. Additionally, let $\pi_{con}(\{y_j\}_{j=1}^M|x; c)$ be the probability of the context-aware model $\pi_{con}$ generating the candidate sequences $\{y_j\}_{j=1}^M$, given the context $c$ and the user query $x$.

First, note that we can write the density of $\pi_{\hat{r},M}$ as follows:

$$\pi_{\hat{r},M}(y|x) = \sum_{\{y_j\}_{j=1}^M \in Y^M} \pi_{\hat{r},M}(y|x, \{y_j\}_{j=1}^M) \pi_{con}(\{y_j\}_{j=1}^M|x; c)$$

$$= \mathbb{E}_{\{y_j\}_{j=1}^M \sim \pi_{con}(\cdot|x;c)} \left[ \pi_{\hat{r},M}(y|x, \{y_j\}_{j=1}^M) \right]$$

$$= \mathbb{E}_{\{y_j\}_{j=1}^M \sim \pi_{con}(\cdot|x;c)} \left[ \sum_j \mathbb{I}\{y_j = y\} \frac{\exp\left(\frac{\hat{r}(x,y_j)}{\tau} + \log \frac{\pi_{ref}(y_j|x)}{\pi_{con}(y_j|x;c)}\right)}{\sum_{\zeta \in \{y_j\}_{j=1}^M} \exp\left(\frac{\hat{r}(x,\zeta)}{\tau} + \log \frac{\pi_{ref}(\zeta|x)}{\pi_{con}(\zeta|x;c)}\right)} \right]$$

$$= \mathbb{E}_{\{y_j\}_{j=1}^M \sim \pi_{con}(\cdot|x;c)} \left[ \sum_j \mathbb{I}\{y_j = y\} \frac{\frac{\pi_{ref}(y_j|x)}{\pi_{con}(y_j|x;c)} \exp\left(\frac{\hat{r}(x,y_j)}{\tau}\right)}{\sum_{\zeta \in \{y_j\}_{j=1}^M} \frac{\pi_{ref}(\zeta|x)}{\pi_{con}(\zeta|x;c)} \exp\left(\frac{\hat{r}(x,\zeta)}{\tau}\right)} \right]$$

$$= \mathbb{E}_{\{y_j\}_{j=1}^M \sim \pi_{con}(\cdot|x;c)} \left[ \frac{\sum_j \mathbb{I}\{y_j = y\}}{\sum_{\zeta \in \{y_j\}_{j=1}^M} \frac{\pi_{ref}(\zeta|x)}{\pi_{con}(\zeta|x;c)} \exp\left(\frac{\hat{r}(x,\zeta)}{\tau}\right)} \right] \frac{\pi_{ref}(y|x)}{\pi_{con}(y|x;c)} \exp\left(\frac{\hat{r}(x,y)}{\tau}\right)$$

$$= \mathbb{E}_{\{y_j\}_{j=1}^M \sim \pi_{con}(\cdot|x;c)} \left[ \frac{\frac{1}{M} \sum_j \mathbb{I}\{y_j = y\}}{\frac{1}{M} \sum_{\zeta \in \{y_j\}_{j=1}^M} \frac{\pi_{ref}(\zeta|x)}{\pi_{con}(\zeta|x;c)} \exp\left(\frac{\hat{r}(x,\zeta)}{\tau}\right)} \right] \frac{\pi_{ref}(y|x)}{\pi_{con}(y|x;c)} \exp\left(\frac{\hat{r}(x,y)}{\tau}\right)$$

Taking the limit as $M \to \infty$ and using the Law of Large Numbers (the sequences are i.i.d.):

$$\lim_{M\to\infty} \pi_{\hat{r},M}(y|x) = \lim_{M\to\infty} \mathbb{E}_{\{y_j\}_{j=1}^M \sim \pi_{\text{con}}(\cdot|x;c)} \left[ \frac{\pi_{\text{con}}(y|x;c)}{\mathbb{E}_{\zeta\sim\pi_{\text{con}}(\cdot|x)}\left[\exp\left(\frac{\hat{r}(x,\zeta)}{\tau}\right)\frac{\pi_{\text{ref}}(\zeta|x)}{\pi_{\text{con}}(\zeta|x;c)}\right]} \right] \frac{\pi_{\text{ref}}(y|x)}{\pi_{\text{con}}(y|x;c)} \exp\left(\frac{\hat{r}(x,y)}{\tau}\right)$$

$$= \pi_{\text{con}}(y|x;c) \frac{1}{\mathbb{E}_{\zeta\sim\pi_{\text{con}}(\cdot|x)}\left[\exp\left(\frac{\hat{r}(x,\zeta)}{\tau}\right)\frac{\pi_{\text{ref}}(\zeta|x)}{\pi_{\text{con}}(\zeta|x)}\right]} \frac{\pi_{\text{ref}}(y|x)}{\pi_{\text{con}}(y|x;c)} \exp\left(\frac{\hat{r}(x,y)}{\tau}\right)$$

$$= \pi_{\text{ref}}(y|x) \frac{1}{\sum_{\zeta\in Y} \pi_{\text{con}}(\zeta|x)\exp\left(\frac{\hat{r}(x,\zeta)}{\tau}\right)\frac{\pi_{\text{ref}}(\zeta|x)}{\pi_{\text{con}}(\zeta|x;c)}} \exp\left(\frac{\hat{r}(x,y)}{\tau}\right)$$

$$= \pi_{\text{ref}}(y|x) \frac{1}{\sum_{\zeta\in Y} \pi_{\text{ref}}(\zeta|x)\exp\left(\frac{\hat{r}(x,\zeta)}{\tau}\right)} \exp\left(\frac{\hat{r}(x,y)}{\tau}\right)$$

$$= \pi_{\text{ref}}(y|x) \frac{1}{\mathbb{E}_{\zeta\sim\pi_{\text{ref}}(\cdot|x)}\left[\exp(\hat{r}(x,\zeta)/\tau)\right]} \exp\left(\frac{\hat{r}(x,y)}{\tau}\right)$$

$\square$

*Corollary 4.1 Proof.* The proof follows directly from Theorem B.1 and Appendix A.1 in Rafailov et al. [2023]. $\square$

## B.3 Formal Connection To Importance Sampling

Given truthful reports by the advertisers, there is a strong connection between our allocation rule and importance sampling. The optimal LLM for the platform's objective $\pi_r^*$ can be interpreted as the target distribution that our allocation rule is trying to simulate. The aim of our allocation rule is to return a reply as similar as possible to a reply drawn from the optimal LLM $\pi_r^*$. This can be interpreted as drawing a reply from $\pi_r^*$ and then evaluating it with respect to the same function, i.e., our allocation rule is trying to estimate $\mathbb{E}_{y\sim\pi_r^*(\cdot|x)}[\pi_r^*(y|x)]$. The LLM $\pi_{\text{gen}}$ that we use to generate the candidate replies (either the reference LLM or the context-aware LLM) can be interpreted as the proposal distribution $\pi_{\text{prop}}(\cdot|x)$ used to generate samples. We are interested in how well, with respect to the target probability $\pi_r^*(\cdot|x)$, a sample $y$ from our estimator using $M$ candidate replies matches a sample drawn from the target distribution $\pi_r^*(\cdot|x)$. In this section, we will analytically show how the quality of the proposal distribution $\pi_{\text{prop}} = \pi_{\text{gen}}$ affects the quality of that estimator.

First, in Lemma B.2 we establish the variance of our estimator:

**Lemma B.2.** *For any LLM $\pi_{gen}$ such that $\pi_{ref}$ is absolutely continuous with respect to $\pi_{gen}$, the variance of the policy $\pi_{r,M}(\cdot|x)$ as an estimator for $\mathbb{E}_{y\sim\pi_r^*(\cdot|x)}[\pi_r^*(y|x)]$ is*

$$Var(\hat{\mu}_{IS}) = \frac{1}{M} \left( \sum_{y\in Y} \frac{\pi_r^*(y|x)^4}{\pi_{gen}(y|x)} - \left( \sum_{y\in Y} \pi_r^*(y|x)^2 \right)^2 \right). \tag{4}$$

*Proof.* Let $y_1, y_2, \ldots, y_M$ be the $M$ generated candidate replies drawn from the proposal distribution $\pi_{\text{prop}}(\cdot|x)$. Conditioned on those candidate replies, the importance sampling estimator for the expected value of the function $\pi_r^*(y|x)$ under the target distribution $\pi_r^*(y|x)$ is

$$\hat{\mu}_{\text{IS}} = \frac{1}{M} \sum_{j=1}^M \pi_r^*(y_j|x) \cdot w(x_j)$$

$$= \frac{1}{M} \sum_{j=1}^M \pi_r^*(y_j|x) \cdot \frac{\pi_r^*(y_j|x)}{\pi_{\text{prop}}(y_j|x)}$$

$$= \frac{1}{M} \sum_{j=1}^M \frac{\pi_r^*(y_j|x)^2}{\pi_{\text{prop}}(y_j|x)}$$

Taking expectation over the generated candidate replies $y_1, \ldots, y_M$:

$$\mathbb{E}[\hat{\mu}_{\text{IS}}] = \mathbb{E}_{y_1,\ldots,y_M \sim \pi_{\text{prop}}(\cdot|x)} \left[ \frac{1}{M} \sum_{j=1}^{M} \frac{\pi_r^*(y_j|x)^2}{\pi_{\text{prop}}(y_j|x)} \right]$$

$$= \frac{1}{M} \sum_{j=1}^{m} \mathbb{E}_{y_j \sim \pi_{\text{prop}}(\cdot|x)} \left[ \frac{\pi_r^*(y_j|x)^2}{\pi_{\text{prop}}(y_j|x)} \right]$$

$$= \mathbb{E}_{y \sim \pi_{\text{prop}}(\cdot|x)} \left[ \frac{\pi_r^*(y|x)^2}{\pi_{\text{prop}}(y|x)} \right]$$

$$= \sum_{y \in Y} \frac{\pi_r^*(y|x)^2}{\pi_{\text{prop}}(y|x)} \pi_{\text{prop}}(y|x)$$

$$= \sum_{y \in Y} \pi_r^*(y|x)^2 \tag{5}$$

Thus, our estimator is unbiased, as expected. The variance of the estimator $\hat{\mu}_{\text{IS}}$ is given by:

$$\text{Var}(\hat{\mu}_{\text{IS}}) = \mathbb{E}[\hat{\mu}_{\text{IS}}^2] - \mathbb{E}[\hat{\mu}_{\text{IS}}]^2 \tag{6}$$

For the term $\hat{\mu}_{\text{IS}}^2$ we have:

$$\hat{\mu}_{\text{IS}}^2 = \left( \frac{1}{M} \sum_{j=1}^{M} \frac{\pi_r^*(y_j|x)^2}{\pi_{\text{prop}}(y_j|x)} \right)$$

$$= \frac{1}{M^2} \sum_{j=1}^{M} \sum_{j'=1}^{M} \frac{\pi_r^*(y_j|x)^2}{\pi_{\text{prop}}(y_j|x)} \cdot \frac{\pi_r^*(y_{j'}|x)^2}{\pi_{\text{prop}}(y_{j'}|x)}$$

Taking the expectation $\mathbb{E}_{\pi_{\text{prop}}(\cdot|x)}[\cdot]$:

$$\mathbb{E}[\hat{\mu}_{\text{IS}}^2] = \frac{1}{M^2} \sum_{j=1}^{M} \sum_{j'=1}^{M} \mathbb{E}_{\pi_{\text{prop}}(\cdot|x)} \left[ \frac{\pi_r^*(y_j|x)^2}{\pi_{\text{prop}}(y_j|x)} \cdot \frac{\pi_r^*(y_{j'}|x)^2}{\pi_{\text{prop}}(y_{j'}|x)} \right]$$

$$= \frac{1}{M^2} \left( \sum_{j=1}^{M} \mathbb{E}_{\pi_{\text{prop}}(\cdot|x)} \left[ \frac{\pi_r^*(y_j|x)^4}{\pi_{\text{prop}}(y_j|x)^2} \right] + \sum_{j \neq j'} \mathbb{E}_{\pi_{\text{prop}}(\cdot|x)} \left[ \frac{\pi_r^*(y_j|x)^2}{\pi_{\text{prop}}(y_j|x)} \right] \right)$$

$$= \frac{1}{M^2} \left( M \sum_{y \in Y} \frac{\pi_r^*(y|x)^4}{\pi_{\text{prop}}(y|x)} + M(M-1) \left( \sum_{y \in Y} \pi_r^*(y|x)^2 \right)^2 \right)$$

$$= \frac{1}{M} \sum_{y \in Y} \frac{\pi_r^*(y|x)^4}{\pi_{\text{prop}}(y|x)} + \frac{M-1}{M} \left( \sum_{y \in Y} \pi_r^*(y|x)^2 \right)^2 \tag{7}$$

The final expression for the variance of the importance sampling estimator with $M$ samples can be computed by substituting Equations (5) and (7) in Equation (6):

$$\text{Var}(\hat{\mu}_{\text{IS}}) = \frac{1}{M} \sum_{y \in Y} \frac{\pi_r^*(y|x)^4}{\pi_{\text{prop}}(y|x)} + \frac{M-1}{M} \left( \sum_{y \in Y} \pi_r^*(y|x)^2 \right)^2 - \left( \sum_{y \in Y} \pi_r^*(y|x)^2 \right)^2$$

$$= \frac{1}{M} \left( \sum_{y \in Y} \frac{\pi_r^*(y|x)^4}{\pi_{\text{prop}}(y|x)} - \left( \sum_{y \in Y} \pi_r^*(y|x)^2 \right)^2 \right) \tag{8}$$

$$\square$$

Lemma B.2 reveals two important details about our estimator. First, the variance of the estimator is inversely proportional to the number of candidate replies generated $M$. Thus, as $M$ increases the variance decreases and our estimator becomes more stable in simulating the target distribution $\pi_r^*(\cdot|x)$. Second, the closer that the proposal distribution $\pi_{\text{prop}}(\cdot|x)$ is to the target distribution, the smaller the term $\sum_{y \in Y} \frac{\pi_r^*(y|x)^4}{\pi_{\text{prop}}(y|x)}$ will be, which reduces the variance of our estimator.

Lemma B.3 establishes how a smaller variance leads to faster convergence:

**Lemma B.3.** *Let* $\mu_{true} = \mathbb{E}_{y \sim \pi_r^*(\cdot|x)}\left[\pi_r^*(y \mid x)\right]$, *and* $\hat{\mu}_{IS}$ *be the importance sampling estimator as in Lemma B.2 using* $M = O\left(\frac{1}{\delta \varepsilon^2} \sum_{y \in Y} \frac{\pi_r^*(y|x)^4}{\pi_{prop}(y|x)}\right)$ *candidate replies. Then, for any LLM* $\pi_{gen}$ *such that* $\pi_{ref}$ *is absolutely continuous with respect to* $\pi_{gen}$, *we have that* $|\hat{\mu}_{IS} - \mu_{true}| < \varepsilon$ *with probability at least* $1 - \delta$.

*Proof.* Let $\hat{\mu}_{\text{IS}}$ be the importance-weighted estimator whose variance is given by Lemma B.2 as

$$\text{Var}(\hat{\mu}_{\text{IS}}) = \frac{1}{M}\left(\sum_{y \in Y} \frac{\pi_r^*(y \mid x)^4}{\pi_{\text{prop}}(y \mid x)} - \left(\sum_{y \in Y} \pi_r^*(y \mid x)^2\right)^2\right).$$

We have already shown in Lemma B.2 that the estimator is unbiased. By Chebyshev's inequality, for any $\varepsilon > 0$,

$$\Pr\left[\left|\hat{\mu}_{\text{IS}} - \mu_{\text{true}}\right| \geq \varepsilon\right] \leq \frac{\text{Var}(\hat{\mu}_{\text{IS}})}{\varepsilon^2} = \frac{1}{M \cdot \varepsilon^2}\left(\sum_{y \in Y} \frac{\pi_r^*(y \mid x)^4}{\pi_{\text{prop}}(y \mid x)} - \left(\sum_{y \in Y} \pi_r^*(y \mid x)^2\right)^2\right)$$

where $\mu_{\text{true}} = \mathbb{E}_{y \sim \pi_r^*(\cdot|x)}\left[\pi_r^*(y \mid x)\right] = \sum_{y \in Y} \pi_r^*(y|x)^2$.

Setting $\Pr[|\hat{\mu}_{\text{IS}} - \mu_{\text{true}}| \geq \varepsilon] < \delta$ and solving for $M$ yields:

$$M = \frac{1}{\delta \, \varepsilon^2}\left(\sum_{y \in Y} \frac{\pi_r^*(y \mid x)^4}{\pi_{\text{prop}}(y \mid x)} - \left(\sum_{y \in Y} \pi_r^*(y \mid x)^2\right)^2\right) = O\left(\frac{1}{\delta \varepsilon^2} \sum_{y \in Y} \frac{\pi_r^*(y \mid x)^4}{\pi_{\text{prop}}(y \mid x)}\right).$$

$\square$

Based on Lemma B.3, our allocation rule converges to the quantity is is trying to approximate at a rate of $\sqrt{M}$.

## B.4 Bounding the Total Variation Distance

In Appendix B.2 we proved that MOSAIC converges in the limit to the theoretically optimal distribution for the platform's objective. Then, in Appendix B.3 we showed how MOSAIC's allocation rule relates to importance sampling, and what that implies for the similarity of MOSAIC's output to that of the optimal distribution $\pi_r^*$ Now we will also show that MOSAIC's probability estimates for any possible sequence converge to that of $\pi_r^*$ at a rate of $\sqrt{M}$

**Theorem B.4.** *Let* $\mathcal{Y}$ *be the domain consisting of all sequences in a finite token vocabulary* $T$ *of length up to* $L$. *Hence* $|\mathcal{Y}| < \infty$. *Assume* $0 \leq w(y) \leq C$ *for all* $y \in \mathcal{Y}$, *where*

$$w(y) = \frac{\pi_r^*(y \mid x)}{\pi_{prop}(y \mid x)}.$$

*Consider* $M$ *i.i.d. draws* $y_1, \ldots, y_M \sim \pi_{prop}(\cdot \mid x)$ *and define the self-normalized importance-sampled distribution induced by Algorithm 1:*

$$\hat{\pi}_{r,M}(y \mid x) = \frac{\sum_{j=1}^{M} w(y_j) \mathbf{1}\{y_j = y\}}{\sum_{j=1}^{M} w(y_j)}.$$

*Then, for any $\varepsilon > 0$ and $\delta > 0$, if $M = O\left(\frac{C^2 \cdot |\mathcal{Y}|^2}{\varepsilon^2} \ln\left(\frac{|\mathcal{Y}|}{\delta}\right)\right)$, with probability at least $1 - \delta$ (over the random draws), the Total Variation distance between $\hat{\pi}_M(\cdot \mid x)$ and $\pi_r^*(\cdot \mid x)$ satisfies $d_{\mathrm{TV}}(\hat{\pi}_M, \pi_r^*) \le \varepsilon$.*

*Proof.* First, we will show that for any $M \in \mathbb{N}$, MOSAIC is an unbiased estimator of $\pi_r^*$.

For each fixed $y \in \mathcal{Y}$, define $X_j^{(y)} := w(y_j)\,\mathbf{1}\{y_j = y\}$ to be a random variable that equals $w(y_j)$ if the $j - th$ draw from the proposal distribution is $y$. Intuitively, this is a weighted indicator for whether the $j$-th draw is equal to $y$.

For any $y \in \mathcal{Y}$, the expected value of this indicator, with the expectation taken over the draw of $y_j \sim \pi_{\mathrm{prop}}$

$$\mathbb{E}\big[X_j^{(y)}\big] \;=\; \sum_{z \in \mathcal{Y}} w(z)\,\mathbf{1}\{z = y\}\,\pi_{\mathrm{prop}}(z) \;=\; w(y)\,\pi_{\mathrm{prop}}(y) \;=\; \pi_r(y).$$

Thus, $\hat{Q}_M(\{y\}) = \frac{1}{M}\sum_{j=1}^M X_j^{(y)}$ is an unbiased estimator of $\pi_r^*(y \mid x)$.

Similarly, $\hat{Q}_M(\mathcal{Y}) = \frac{1}{M}\sum_{j=1}^M w(y_j)$ is an unbiased estimator of 1 (since $\mathbb{E}[w(y_j)] = 1$) (assuming that $\pi_r^*$ is the optimal policy, i.e., a normalized LLM so that the probabilities sum up to one).

The self-normalized distribution can be written as

$$\hat{\pi}_{r,M}(y \mid x) \;=\; \frac{\hat{Q}_M(\{y\})}{\hat{Q}_M(\mathcal{Y})}.$$

*Hoeffding's inequality* for bounded random variables tells us that for any $\varepsilon' > 0$:

$$\Pr\Big[\big|\hat{Q}_M(\{y\}) - \pi_r^*(y \mid x)\big| \;\ge\; \varepsilon'\Big] \;\le\; 2\exp\Big(-\frac{2\,M\,\varepsilon'^2}{C^2}\Big).$$

For each $y \in \mathcal{Y}$, we have the same deviation event, so by a union bound over all $y \in \mathcal{Y}$:

$$\Pr\Big[\exists\, y \in \mathcal{Y}:\; \big|\hat{Q}_M(\{y\}) - \pi_r^*(y \mid x)\big| \;\ge\; \varepsilon'\Big] \;\le\; \sum_{y \in \mathcal{Y}} 2\exp\Big(-\frac{2M\varepsilon'^2}{C^2}\Big) \;=\; 2\,|\mathcal{Y}|\,\exp\Big(-\frac{2M\varepsilon'^2}{C^2}\Big) \le \delta_1.$$

Thus, using $M = O\left(\frac{C^2}{\varepsilon'^2} \ln\left(\frac{|\mathcal{Y}|}{\delta_1}\right)\right)$ candidate replies, with probability at least $1 - \delta_1$ over the draw of those replies, we have *simultaneously* for all $y \in \mathcal{Y}$, $|\hat{Q}_M(\{y\}) - \pi_r^*(y \mid x)| \le \varepsilon'$.

By a similar argument, applying Hoeffding's inequality to $\hat{Q}_M(\mathcal{Y})$, which is an unbiased estimator of 1:

$$\Pr\Big[\big|\hat{Q}_M(\mathcal{Y}) - 1\big| \;\ge\; \varepsilon'\Big] \;\le\; 2\,\exp\Big(-\frac{2M\,\varepsilon'^2}{C^2}\Big).$$

Hence, with high probability $1 - \delta_2$, $\hat{Q}_M(\mathcal{Y})$ is also within $\pm\varepsilon$ of 1. A union bound over both events (all $y \in \mathcal{Y}$ plus $\hat{Q}_M(\mathcal{Y})$) yields a final event $E_\varepsilon'$ with probability $1 - (\delta_1 + \delta_2)$ on which:

$$\hat{Q}_M(\{y\}) \;\in\; [\pi_r^*(y) - \varepsilon',\, \pi_r^*(y) + \varepsilon'], \quad \hat{Q}_M(\mathcal{Y}) \;\in\; [1 - \varepsilon',\, 1 + \varepsilon'], \quad \forall\, y \in \mathcal{Y}.$$

On this good event $E_{\varepsilon'}$, for each $y \in \mathcal{Y}$:

$$\hat{\pi}_M(y|x) \;=\; \frac{\hat{Q}_M(\{y\})}{\hat{Q}_M(\mathcal{Y})} \;\in\; \left[\frac{\pi_r^*(y|x) - \varepsilon'}{1 + \varepsilon},\, \frac{\pi_r^*(y|x) + \varepsilon'}{1 - \varepsilon'}\right]$$

Recall the total variation (TV) distance between two discrete distributions $P$ and $Q$ on $\mathcal{Y}$ is

$$d_{\mathrm{TV}}(P,\, Q) \;=\; \frac{1}{2}\sum_{y \in \mathcal{Y}} \big|P(y) - Q(y)\big|.$$

We aim to show that on the event $E'_\varepsilon$ (which holds with probability at least $1 - (\delta_1 + \delta_2)$), the quantity

$$\sum_{y \in \mathcal{Y}} \left| \hat{\pi}_M(y|x) - \pi_r^*(y|x) \right|$$

is at most $O(\varepsilon')$.

On $E_{\varepsilon'}$, we have, for each $y \in \mathcal{Y}$,

$$\hat{\pi}_M(y|x) = \frac{\hat{Q}_M(\{y\})}{\hat{Q}_M(\mathcal{Y})} \in \left[ \frac{\pi_r^*(y|x) - \varepsilon'}{1 + \varepsilon'}, \ \frac{\pi_r^*(y|x) + \varepsilon'}{1 - \varepsilon'} \right].$$

Thus:

$$\hat{\pi}_M(y|x) - \pi_r^*(y|x) \in \left[ -\frac{\varepsilon' \pi_r^*(y|x)}{1 + \varepsilon'} - \frac{\varepsilon'}{1 + \varepsilon'}, \ \frac{\varepsilon' \pi_r^*(y|x)}{1 - \varepsilon'} + \frac{\varepsilon'}{1 - \varepsilon'} \right] \tag{9}$$

$$\in \left[ -\frac{2\varepsilon}{1 + \varepsilon}, \ \frac{2\varepsilon}{1 - \varepsilon} \right] \tag{10}$$

Thus, conditioned on the good event $E_{\varepsilon'}$, we have $\left| \hat{\pi}_M(y) - \pi_r^*(y) \right| = O(\varepsilon')$ by a Taylor expansion for small $\varepsilon'$.

Therefore, on the event $E'_\varepsilon$,

$$d_{\mathrm{TV}}\left( \hat{\pi}_M, \pi_r^* \right) = \frac{1}{2} \sum_{y \in \mathcal{Y}} \left| \hat{\pi}_M(y) - \pi_r^*(y) \right| \leq \frac{1}{2} \sum_{y \in \mathcal{Y}} O(\varepsilon') = O\left( |\mathcal{Y}| \cdot \varepsilon' \right).$$

Setting $\varepsilon' = \frac{\varepsilon}{|\mathcal{Y}|}$ to achieve the target error $\varepsilon$ concludes the proof.

$\square$

# C  Details from Section 5

In this section, we present all omitted details from Section 5.

## C.1  Omitted Proofs from Section 5.1

*Theorem 5.1 Proof.* Let $\hat{\mathbf{r}}_i = (\hat{r}_i(x, y_1), \ldots, \hat{r}_i(x, y_M))$ be the reward reports of advertiser $i$ for the $M$ generated candidate sequences. Then, for MOSAIC's allocation rule, holding the candidate sequences and the reports of all other advertisers fixed, the probability of returning each of the $M$ candidate sequences as a function of $i$'s reports is:

$$\pi(\mathbf{r}_i; \vec{\beta}_{-i}) = \text{softmax}\left(\frac{\mathbf{r}_i}{\tau} + \beta_{-i}\right), \tag{11}$$

where $\boldsymbol{\beta}_{-i,j} = \frac{\sum_{k \in N \setminus \{i\}} \hat{r}_k(x, y_j)}{\tau} + \log \frac{\pi_{\text{ref}}(y_j|x)}{\pi_{\text{con}}(y_j|x;c)}$. Importantly $\boldsymbol{\beta}_{-i}$ is an $M$-dimensional vector that does not depend on advertiser $i$'s reports.

We would like to equip $\pi(\cdot, \boldsymbol{\beta}_{-i})$ with a payment rule $p(\cdot; \boldsymbol{\beta}_{-i})$ so that the resulting mechanism $(\pi(\cdot, \boldsymbol{\beta}_{-i}), p(\cdot; \boldsymbol{\beta}_{-i}))$ will be strategyproof. This requires that $\pi(\cdot, \boldsymbol{\beta}_{-i})$ have a property known as *cyclic monotonicity*. Equivalently, $\pi(\cdot, \boldsymbol{\beta}_{-i})$ must be the (sub)gradient of advertiser $i$'s utility for bidding truthfully in the mechanism $U(\boldsymbol{r}_i; \boldsymbol{\beta}_{-i})$, and that utility function must be convex [Frongillo and Kash, 2021, Rochet, 1987, Myerson, 1981].

It is easy to verify that for the function class:

$$U_C(\boldsymbol{r}_i; \boldsymbol{\beta}_{-i}) = \tau \log\left(\sum_{j=1}^{M} \exp\left(\frac{r_i(x, y_j)}{\tau} + \boldsymbol{\beta}_{-i,j}\right)\right) + C, \ C \in \mathbb{R} \tag{12}$$

the allocation rule $\pi(\boldsymbol{r}_i; \boldsymbol{\beta}_{-i})$ is a gradient of $U_C(\boldsymbol{r}_i; \boldsymbol{\beta}_{-i})$. Additionally, $U_C(\boldsymbol{r}_i; \boldsymbol{\beta}_{-i})$ is convex in $\boldsymbol{r}_i$: the exponential function $e^x$ is (strictly) convex, because its second derivative is positive. The transformation $\frac{r_i(x, y_j)}{\tau} + \boldsymbol{\beta}_{-i,j}$ is an affine transformation of $r_i(x, y_j)$, and affine transformations preserve convexity. Finally, it is well-known that the LogSumExp function is convex.

Thus, for any $\boldsymbol{\beta}_{-i}$ and for any set of generated candidate sequences, reporting truthfully maximizes advertiser $i$'s expected utility, with the expectation taken over the draw of the final sequence from the set of candidate sequences. Adopting the quasi-linear utility model, advertiser $i$'s payment is:

$$U_C(\boldsymbol{r}_i; \boldsymbol{\beta}_{-i}) = \pi(\boldsymbol{r}_i; \boldsymbol{\beta}_{-i}) \cdot \boldsymbol{r}_i - p(\boldsymbol{r}_i; \boldsymbol{\beta}_{-i})$$
$$p(\boldsymbol{r}_i; \boldsymbol{\beta}_{-i}) = \pi(\boldsymbol{r}_i; \boldsymbol{\beta}_{-i}) \cdot \boldsymbol{r}_i - U_C(\boldsymbol{r}_i; \boldsymbol{\beta}_{-i})$$

$$p(\boldsymbol{r}_i; \boldsymbol{\beta}_{-i}) = \pi(\boldsymbol{r}_i; \boldsymbol{\beta}_{-i}) \cdot \boldsymbol{r}_i - \tau \log\left(\sum_{j=1}^{M} \exp\left(\frac{r_i(x, y_j)}{\tau} + \boldsymbol{\beta}_{-i,j}\right)\right) - C, \ C \in \mathbb{R} \tag{13}$$

$\square$

## C.2 Differences from Standard Auction Settings

Standard auction environments typically rely on a set of assumptions that simplify mechanism design; however, these assumptions do not apply to auctions for LLM-generated content. In this section, we detail these assumptions and discuss why they are inapplicable in our context.

First, in a standard auction setting, it is common to assume that the agents' valuation functions satisfy free disposal, i.e., $v_i(S) \geq v_i(S')\ \forall S \supseteq S', S, S' \supseteq \mathcal{I}$. The interpretation of free disposal is that an agent can discard any items she is allocated that she is not interested in. Free disposal combined with the fact that an agent has zero value for the empty bundle mean that her value for any outcome is weakly positive. Second, in most auction environments, the allocation rule is different for different agents: each agent will get allocated her own bundle of items, and we can assume that she is indifferent to the allocation of items to the other agents.

As detailed in Rafailov et al. [2023], assuming that an agent's LLM $\pi_i$ was trained to maximize her reward function (and regularized with respect to its KL divergence from some reference LLM, which we assume to be the same as the auctioneer's reference LLM), there is a one-to-many mapping between an advertiser's optimal LLM, and her implicit reward function. That mapping is:

$$r_i(x, y) = \tau_i \log \frac{\pi_i(y|x)}{\pi_{\text{ref}}(y|x)} + \log Z_i(x) \tag{14}$$

where $Z_i(x)$ is a prompt-dependent constant, and $\tau_i$ is the regularization hyperparameter of advertiser $i$, similar to the one in Equation (1). All functions in the class defined in Equation (14) are equivalent, in the sense that they induce exactly the same LLM [Rafailov et al., 2023]. This has two implications: First, unlike standard auction environments, an agent's reward can go negative – there is nothing equivalent to the free disposal property. Setting $Z_i(x)$ to zero (which is equivalent to normalizing the induced probabilities by the LLM [Rafailov et al., 2023]), the agent's reward is negative for any sequence for which her LLM assigns a lower probability than $\pi_{\text{ref}}$.

Second, especially in the online advertising application, an agent's expected utility for not participating in the auction is negative: if advertiser $i$ does not participate in the auction, her payment is zero, but her expected value for the outcome is

$$\pi(\mathbf{0}; \boldsymbol{\beta}_{-i}) \cdot \boldsymbol{r}_i = \pi(\boldsymbol{\beta}_{-i}) \cdot \boldsymbol{r}_i \tag{15}$$

The other advertisers have very low rewards for the sequences that mention advertiser $i$: assuming their LLMs have been properly trained, they will evaluate all sequences that explicitly mention a different, possibly competing brand, as unlikely. Thus, based on Equation (14) the corresponding advertisers have very low rewards for those sequences and conversely, advertiser $i$ has low rewards for the sequences that the other advertisers have high rewards for. But based on Equation (2), if advertiser $i$ does not participate in the auction, $\pi(\mathbf{0}; \boldsymbol{\beta}_{-i})$ will assign high probabilities to sequences for which $i$ has low rewards for. Thus, Equation (15) implies that, unlike standard auction environments, the advertiser's expected reward and utility for not participating in the mechanism is negative.

## C.3 Our mechanism is "almost individually rational"

First, we explain why the standard notion of individual rationality (i.e., weakly positive utility from participation in the mechanism) encountered in most auction settings is impossible to achieve in this domain while converging to the optimal distribution and maintaining incentive compatibility. Then, we explain how, with our payment offset, our mechanism is "almost IR:" In Lemma C.1 we prove that the ex-interim utility of an advertiser who has zero reward for all candidate sequences and bids truthfully is deterministically zero, i.e., advertisers that do not contribute to the social welfare (but also do not detract from it) have zero utility. Similarly, in Lemma C.2 we prove that if an agent's reward for all candidate sequences is (weakly) positive, then her ex-interim utility is (weakly) positive.

**Why is individual rationality (IR) impossible?** *Individual rationality* (IR) stipulates that an agent gains more utility by participating and bidding truthfully in a mechanism than by not participating at all. Typically, if an agent's utility for non-participation is zero, participating should yield weakly positive utility. However, this simplification does not apply in our setting.

As discussed in Section 5.2, advertiser $i$'s reward for any sequence $y$ can be arbitrarily negative (Equation (14)). The same is true for the utility from truthful participation, as outlined in Equation (12). To ensure a positive utility for every advertiser in our mechanism, an offset would need to be infinitely large or dependent on advertiser $i$'s reports. But then the mechanism's allocation rule would no longer be the gradient of advertiser $i$'s utility with respect to her reports, which would destroy strategyproofness [Frongillo and Kash, 2021, Rochet, 1987, Myerson, 1981].

It is important to note that this challenge is inherent not just to our mechanism but to any mechanism in this setting that operates with a fixed set of sequences, aims to approximate the optimal distribution, and maintains strategyproofness. Under these conditions, the only allocation rule that approximates the theoretically optimal distribution (Equation (2)) is that of our mechanism. However, this uniquely determines the advertisers' utilities, up to a constant factor, as described in Equation (12) [Frongillo and Kash, 2021, Rochet, 1987, Myerson, 1981].

**Lemma C.1.** *For the payment offset $C = -\tau \log \left( \sum_{j=1}^{M} \exp \left( \boldsymbol{\beta}_{-i,j} \right) \right)$ if advertiser $i$'s reward for all candidate sequences is zero, then her ex-interim utility is deterministically zero, for all $\boldsymbol{\beta}_{-i} \in \hat{R}_{-i}$.*

*Lemma C.1 Proof.* First, note that for all $\boldsymbol{\beta}_{-i} \in \hat{R}_{-i}$, advertiser $i$'s expected reward for the outcome is zero, as $\pi(\boldsymbol{r}_i; \boldsymbol{\beta}_{-i}) \cdot \boldsymbol{r}_i = \pi(\boldsymbol{r}_i; \boldsymbol{\beta}_{-i}) \cdot \mathbf{0} = 0$. Additionally, advertiser $i$'s reward for the realized outcome will deterministically be zero, as her reward for all generated candidate sequences is zero. Finally, note that by setting $\boldsymbol{r}_i = \mathbf{0}$ in Equation (13) with the offset $C$ set as in Section 5.3, we have that the advertiser $i$'s payment is also deterministically zero. Thus, an advertiser with zero reward for all generated candidate sequences who reports her rewards truthfully has deterministically zero reward for the final outcome and zero payments, and her utility is also deterministically zero.

$\square$

**Lemma C.2.** *For the payment offset $C = -\tau \log \left( \sum_{j=1}^{M} \exp \left( \boldsymbol{\beta}_{-i,j} \right) \right)$ if advertiser $i$'s reward for all candidate sequences is positive, then her ex-interim utility is positive, for all reports $\boldsymbol{\beta}_{-i} \in \hat{R}_{-i}$.*

*Proof.* Lemma C.1 establishes that when advertiser $i$'s reward for all candidate sequences is zero, her utility for truthfully bidding in the mechanism, denoted as $U(\mathbf{0}; \boldsymbol{\beta}_{-i})$, is zero for all possible reports of the other advertisers $\boldsymbol{\beta}_{-i} \in \hat{R}_{-i}$.

Furthermore, Theorem 5.1 shows that the mechanism's allocation rule corresponds to the gradient of advertiser $i$'s utility when bidding truthfully. Because the allocation rule is non-negative, the gradient of advertiser $i$'s utility for bidding truthfully is also non-negative.

Thus, if advertiser $i$'s rewards for all candidate sequences are weakly positive, and considering the non-negative gradient of her utility, her ex-interim utility under truthful bidding must be positive, irrespective of the other advertisers' reports $\boldsymbol{\beta}_{-i}$.

$\square$

**Corollary C.3.** *For the payment offset $C = -\tau \log \left( \sum_{j=1}^{M} \exp \left( \boldsymbol{\beta}_{-i,j} \right) \right)$ if the distribution $\pi_{con}$ only generates candidate sequences for which advertiser $i$'s reward is positive, then the ex-ante expected utility of the advertiser is positive.*

*Corollary C.3 Proof.* This follows immediately from the fact the the fact that the ex-ante utility of the advertiser is the expectation of her ex-interim utility with respect to her reward for the generated sequences, and the fact that the second quantity is positive whenever the reward of the advertiser for all candidate sequences is positive from Lemma C.2.

$\square$

## C.4 "What you give is what you get"

As we explained in Section 5.3.2, our allocation rule, which is the only one over a finite set of replies that converges to the optimal LLM, is also the (sub)gradient of the utility to ensure truthfulness (Rochet, 1987). Because the allocation rule is the same for all advertisers, their utilities must also be the same, up to advertiser-specific offsets, as indicated by Equation (16):

$$U_C(\boldsymbol{r}_i; \boldsymbol{\beta}_{-i}) = \tau \log \left( \sum_{j=1}^{M} \exp \left( \frac{1}{\tau} \sum_{k \in N} r_k(x, y_j) \right) + \log \frac{\pi_{\mathrm{ref}}(y_j|x)}{\pi_{\mathrm{con}}(y_j|x; c)} \right) + C, \ C \in \mathbb{R} \quad (16)$$

However, not all advertisers contribute equally to the social welfare of the final outcome. Because of this, implementing the mechanism without a carefully-designed offset would lead to free-riding: as long as an agent's utility in Equation (16) is positive, she would be incentivized to participate, even if the user query was completely unrelated to her business, because the mechanism would ensure that she received, on expectation, the same (positive) expected utility from doing so as any other participating advertiser.[9]

Incentivizing unrelated advertisers to participate would have adverse effects. First, the better-performing context-aware mechanism would create candidate sequences with worse rewards for *all* advertisers, because its context would be "diluted" from advertisers unrelated to the user query. In our running example for the query "How to learn music online?", imagine adding "Try to mention 'EasySwitch', a comprehensive VPN service" to the context of the context-aware LLM $\pi_{\mathrm{con}}(\cdot|x; c)$.

Additionally, for both versions of the mechanism, following the discussion in Section 5.2, the advertisers for whom the user query is unrelated are more likely to have negative rewards for the generated sequences as their LLMs will deem the candidate sequences more unlikely than the reference LLM. Thus, based on their utility according to Theorem 5.1, their participation in the mechanism will lead to a reduction of the total sum of rewards of the generated sequences for the advertisers, which will indirectly reduce the expected utility of all advertisers, making the mechanism less attractive for the user-query-relevant advertisers.

To summarize, all advertisers receiving the same utility would incentivize advertisers for whom the user query is unrelated to participate in the auction. This would in turn reduce everyone's expected utility, potentially reducing the incentive for the user-query-relevant advertisers to participate, and lead to sequences with worse expected rewards for the advertisers and usefulness for the user.[10] Thus, in the application of auctions for aggregating advertisers' preferences over LLM-generated outputs, advertisers with higher contribution to social welfare also receiving proportionally higher utility by the mechanism is important for the long-term success of the mechanism in practice.

---

[9]We can assume that advertisers can estimate their expected utility from participation using historical data from past auctions, analogously to how they can estimate their utility for participating in sponsored search auctions.

[10]If we interpret the KL divergence between the distribution induced by the reference LLM and the LLM that generated the candidate sequences as a measure of their expected usefulness for the user.

# D  Details from Section 6

## D.1  Detailed Experiment Setup

All synthetic instances are provided in Appendix E. We use Llama-2-7b-chat-hf as the reference LLM [Touvron et al., 2023], which uses the Llama Community License. The context-aware LLM is created as described in Footnote 4.

Following Rafailov et al. [2023], the advertisers' reward functions are defined as $r_i(x, y) = \log \frac{\pi_i(y|x)}{\pi_{\text{ref}}(y|x)}$. For the auctioneer's objective, we set $\tau = 1$ in Equation (1), balancing advertisers' rewards and divergence from the reference LLM.

We use 50 user queries and test each query on 25 different random seeds, resulting in 1,250 instances. Following Li et al. [2024], Rozière et al. [2024], we sample from all LLMs using a temperature of 0.8 and top-p 0.95.

We create a set of synthetic instances to test our mechanism. Each instance consists of a user query, e.g. "How do I bake cookies?" and a list of advertisers. Each advertiser is defined by an "advertiser name", e.g. "KitchenFix" and an advertiser description, e.g., "producing kitchen appliances."[11] The reference LLM $\pi_{\text{ref}}$ responsible for generating replies that are useful for the user is Llama-2-7b-chat-hf [Touvron et al., 2023]. In Appendix D.10, we replicate these experiments using Google's flan-t5-large model [Chung et al., 2022], observing qualitatively very similar results. Following Dütting et al. [2024], we create the advertisers' LLMs by adding advertising instructions to the reference LLM. The advertisers' LLMs are created using the same reference LLM, and adding the instruction: "Answer the question advertising ⟨advertiser⟩, ⟨advertiser description⟩." The context aware LLM is created using the same reference LLM, and adding the instruction: "Answer the query. Try to mention ⟨advertiser 1⟩, who ⟨advertiser description 1⟩  and ⟨advertiser 2⟩, who ⟨advertiser description 2⟩."

Following [Rafailov et al., 2023] the reward function of advertiser $i$ is set to $r_i(x, y) = \log \frac{\pi_i(y|x)}{\pi_{\text{ref}}(y|x)}$, where $\pi_i$ is advertiser $i$'s LLM, i.e., we set $\tau_i = 1, Z_i(x) = 1$ for all advertisers and for all user prompts in Equation (14).[12] For the auctioneer's objective as defined in Equation (1) we set $\tau = 1$, balancing between the advertisers' expected rewards for the generated sequences and the sequences' divergence from the reference LLM responsible for generating useful replies for the user. Thus, the optimal policy according to Equation (2) becomes:

$$\pi_r^*(y|x) = \frac{1}{Z(x)} \pi_{\text{ref}}(y|x) \exp \left( \sum_{i \in N} r_i(x, y) \right) \tag{17}$$

Following Li et al. [2024], Rozière et al. [2024] we sample from the LLM generating the sequences (either $\pi_{\text{ref}}$ or $\pi_{\text{con}}$) with temperature 0.8 and top-p 0.95. We use 50 user queries, each with two interested advertisers. To increase the statistical significance of our results, we test each user query on 25 different random seeds, effectively resulting in 1250 individual instances. The seed affects both the generation of the candidate sequences, and the randomized sampling of the final sequence from the candidate sequences. We will make all of our code and set of synthetic instances publicly available.

All experiments were conducted on a compute cluster running Ubuntu 20.04.6 LTS with AMD EPYC processors with 48 cores and 1512GB RAM and Nvidia A100 GPUs and Python 3.12.1.

## D.2  Choice of baselines

There are no other mechanisms in the literature for aggregating multiple LLMs to compare against. The only other mechanism that aggregates multiple LLMs is Dütting et al. [2024], which is not strategy-proof and does not converge to the platform's objective. The only other potential comparison is VCG, which, as explained in Section 3.2, is not applicable since it does not align with the platform's objective. Hence, the best baseline for comparison is naive MOSAIC, which we consider

---

[11]Following Dütting et al. [2024], we use fictitious brand names to avoid the model using any existing ad.

[12]Note that we cannot estimate the true value of $Z_i(x)$. However, it is a query-dependent constant. Given that our experiments focus on the reward difference of different sequences for the same query, this is a non-issue.

a strong benchmark given that it provably converges to the platform's optimal distribution based on Corollary 4.1.

## D.3 Comparison of baseline and context-aware mechanism outputs

To provide some intuition of how the context-aware mechanism improves the relevance of generated responses for the advertisers, we present a side-by-side comparison of outputs from both mechanisms for a specific query in Table 2. Note that only the context-aware mechanism is able to incorporate the advertisers into the reply to the user's query.

| **User query** |
| --- |
| What are effective ways to learn a musical instrument online? |
| **Advertisers** |
| - MusicMastery: offering online music lessons and virtual tutoring
- InstaTune: selling musical instruments and learning aids |
| **Context-Aware Mechanism Output** |
| Effective ways to learn a musical instrument online include:
1. Online music lessons: Platforms like **MusicMastery** offer one-on-one virtual tutoring, group classes, and pre-recorded lessons. These lessons can be tailored to your skill level and learning style.
2. Interactive learning tools: **InstaTune** offers a range of interactive learning tools, such as apps, software, and games, that make learning fun and engaging.
3. Practice with technology: Utilize digital tools like metronomes, tuners, and recording software to practice and track your progress.
4. Collaborate with other musicians: Join online communities, forums, and social media groups to connect with other musicians and learn from their experiences. . . . |
| **Baseline Mechanism Output** |
| - Practice consistently and regularly: Consistent practice is essential to learning any musical instrument, and it's even more important when learning online. Set aside a specific time each day or week to practice, and make sure to stick to it.
- Find a good teacher or mentor: While it's possible to learn some things on your own, working with a good teacher or mentor can help you improve faster and avoid developing bad habits. Look for someone who is experienced and knowledgeable in the type of music you want to learn.
- Use technology to your advantage: There are many online tools and resources available that can help you learn an instrument. These include video lessons, online tutorials, and apps that can help you practice and track your progress. . . . |

Table 2: Comparison of context-aware and baseline mechanism outputs using 10 candidate sequences. Advertisers highlighted in bold, outputs truncated to conserve space.

## D.4 Experimental Evaluation of the Usefulness to the User

In this section, we experimentally evaluate the usefulness of the replies returned by MOSAIC from the user's perspective.

In Figure 3, we plot the log probability of the replies returned by MOSAIC with respect to the optimal LLM for the user (i.e., the reference LLM) as a function of the number of candidate replies generated. We compare the context-aware version of MOSAIC to a baseline version. To benchmark our results, we also compare against sampling a *single reply* directly from the optimal distribution for this task, which is the reference LLM itself.

The results in Figure 3 show that for the context-aware version of our mechanism, the log probability of the replies relative to the reference LLM increases rapidly as the number of candidate replies grows. This demonstrates that by generating more replies, the context-aware version of MOSAIC is able to produce replies with substantial value for the user. By comparison, the usefulness to the user that the baseline version is able to achieve does not scale with more generated replies.

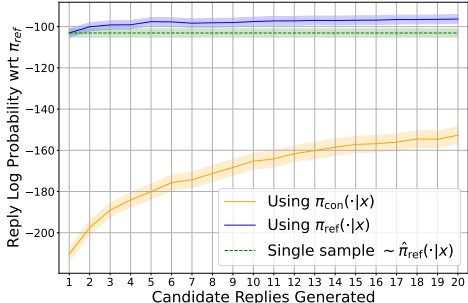

Figure 3: Reply log probability with respect to the reference LLM as a function of the number of replies generated using $\pi_{\mathrm{ref}}$ and $\pi_{\mathrm{con}}$.

However, there remains a gap between the log probability of those replies and the benchmark set by the reference LLM. This discrepancy arises because the platform optimizes a different objective: the expected advertiser rewards combined with the KL divergence from the reference LLM. By increasing the weight $\tau$ that the platform places on the reference LLM, this gap between the context-aware version of MOSAIC and the user's optimal benchmark will close.

## D.5    Comprehensive Experimental Evaluation of the Offset from Section 5.3

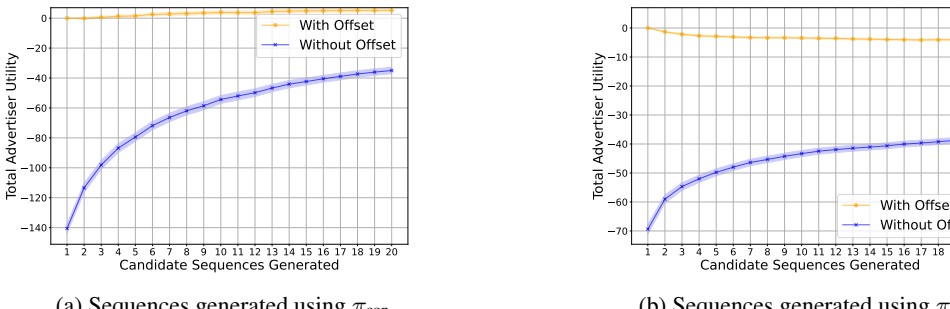

(a) Sequences generated using $\pi_{\mathrm{con}}$         (b) Sequences generated using $\pi_{\mathrm{ref}}$

Figure 4: Comparison of total advertiser utility gain from participation with, and without the payment offset, as a function of the number of candidate sequences generated using $\pi_{\mathrm{ref}}$ and $\pi_{\mathrm{con}}$. Averaged over 1250 runs including 95% CIs.

In this section, we provide a comprehensive experimental evaluation of the payment offset introduced in Section 5.3. Figure 4 explores the effectiveness of the payment offset introduced in Section 5.3 in ensuring that our mechanism is ex-ante IR, i.e., advertisers have positive expected utility gains from participating. To that end, we compare the advertiser utility gain from participation with and without the payment offset, as a function of the number of candidate sequences generated by the context-aware (Figure 4a) and baseline (Figure 4b) versions of our mechanism.

In Figure 4a we observe that for the context-aware version of our mechanism, adding the payment offset introduced of Section 5.3 to the payment rule is enough to ensure positive expected utility for the advertisers (conditioned on the fact that they are related to the user's query), i.e., make the mechanism ex-ante IR. Notably, without the payment offset, advertisers have very negative expected utility from participating in the mechanism.

In Figure 4b we observe that for the baseline version of our mechanism, adding the payment offset introduced of Section 5.3 to the payment rule causes a very large increase in the advertisers' expected utility, but it is still not enough to ensure positive expected utility for the advertisers. To conclude, Figure 4 demonstrates that the offset introduced in Section 5.3 causes a large increase in

the advertisers' expected utility, which in case of the context-aware version of the mechanism, is also enough to make the mechanism ex-ante IR.

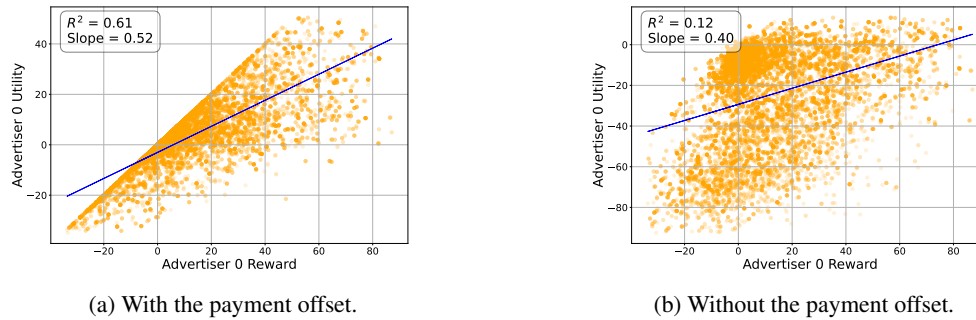

(a) With the payment offset.  (b) Without the payment offset.

Figure 5: Comparative scatter plots of advertiser reward and utility gain from participation, with and without the payment offset of Section 5.3 for candidate sequences generated by the context-aware LLM $\pi_{\text{gen}}$. We additionally show a linear regressor fit to that data, its slope and its $R^2$.

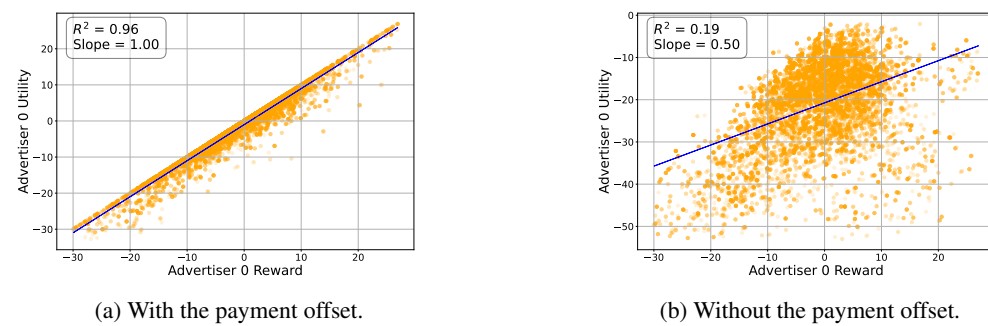

(a) With the payment offset.  (b) Without the payment offset.

Figure 6: Comparative scatter plots of advertiser reward and utility gain from participation, with and without the payment offset of Section 5.3 for candidate sequences generated by the reference $\pi_{\text{ref}}$. We additionally show a linear regressor fit to that data, its slope and its $R^2$.

Figures 5 and 6 explore the effectiveness of the payment offset introduced in Section 5.3 in aligning an advertiser's utility with her contribution to the social welfare. In Figure 5 we compare the scatter plots of the advertiser reward and utility gain from participation in the mechanism, with and without the payment offset introduced in Section 5.3 for candidate sequences generated using the context-aware LLM $\pi_{\text{ref}}$. Additionally, for both subfigures, we show a linear regressor fitted to the data, as well as its slope and coefficient of determination. Comparing the two subfigures, it is immediately obvious that adding the offset to the payments makes the relationship between advertiser reward and utility gain far more linear. This is confirmed by the coefficient of determination of the linear regressors fit to each dataset. The coefficient of determination of the linear regressor is far larger when we use the offset. Without the payment offset, the coefficient of determination is almost 0, indicating that, without our payment offset, reward gain is not a predictive measure of an agent's utility. Additionally, the slope of the linear regressor is also higher for the scatter plot with the payment offset.

In Figure 6 we make the same comparison, but for candidate sequences generated using the reference LLM $\pi_{\text{ref}}$. The results are now even more pronounced. In Figure 6a we observe the relationship between advertiser utility and reward gain with our payment offset is almost perfectly linear, as suggested by the linear regressor fitted to the data having a slope of $1.00$ and an extremely high coefficient of determination of $0.96$, indicating that it can almost perfectly fit the data. Without our payment offset however, in Figure 6b we can see that the relationship between the two metrics is again both less linear, and less positively correlated, as the slope of the linear regressor is $0.5$ and its coefficient of determination is only $0.19$.

To conclude, in all cases tested, the use of the advertiser-specific offset introduced in Section 5.3 increases an advertiser's expected utility, makes the relationship between an advertiser's contribution

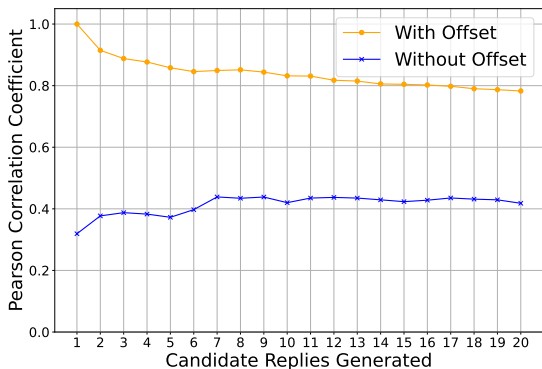

Figure 7: Pearson correlation between advertiser normalized utility and reward.

to social welfare and her utility both more linear and more positively correlated, and, for the context-aware version of our mechanism, can ensure positive expected utility for the advertisers.

### D.6 Experiments in Settings with Many Advertisers

In this section, we test MOSAIC's performance in settings with $n = 5$ and $n = 10$ advertisers, compared to the results of Section 6 where the number of advertisers was set to 2. To maintain the same balance between advertisers and the usefulness to the user as in the experiments of the main paper, we set $\tau = n/2$.[13] We use the same 50 user queries, but now each query has five and ten advertisers, and test each query on 20 different random seeds, resulting in 1000 instances.

In Figures 8a to 8j, we present the log probability of the returned reply with respect to the optimal LLM for the platform's objective $\pi_r^*$ and the reference LLM $\pi_{\text{ref}}$, the total advertiser value and reward gain from participating in MOSAIC and the platform's revenue. All plots are with respect to the number of candidate replies (i.e., LLM queries) that MOSAIC used. We compare MOSAIC's baseline and context-aware versions.

First, in Figures 8a and 8b we observe that the context-aware version of MOSAIC quickly converges to the platform's objective. By comparison, MOSAIC's naive version fails to do so. It is noteworthy that for very low numbers of candidate replies, the naive version of the mechanism performs better than the context-aware version. That is because in these experiments, more weight is placed towards the reference LLM, as we have set a higher $\tau$ value. However, the superior convergence rate of the context-aware version of the mechanism recovers that difference very quickly. At the same time, Figures 8c and 8d show that the context-aware version of the mechanism produces sequences that closely match the distribution of the reference LLM, thus maintaining the usefulness to the user.

Finally, the context-aware version of MOSAIC is able to generate significant value (Figures 8e and 8f) and utility (Figures 8g and 8h) for the advertisers, while also recapturing a significant portion as revenue for the platform (Figures 8i and 8j).

Taken together, similar to our results in the main paper, we have shown that even with a large number of advertisers, MOSAIC quickly converges to the theoretically optimal distribution, generating significant value and utility for the advertisers and revenue for the platform, while also maintaining its usefulness to the user.

---

[13]Note that in the main experiments of Section 6 we had $n = 2$ and $\tau = 1$

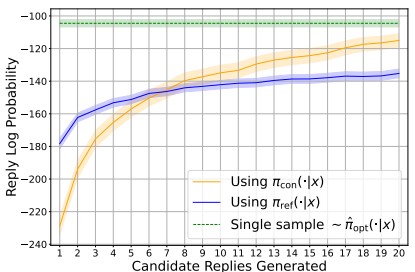

(a) Log probability w.r.t. $\pi_r^*$ for 5 advertisers.

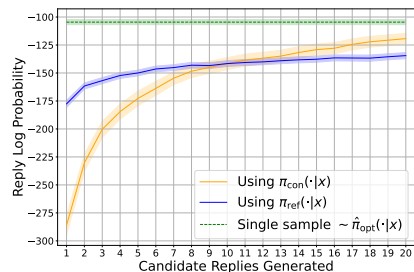

(b) Log probability w.r.t. $\pi_r^*$ for 10 advertisers.

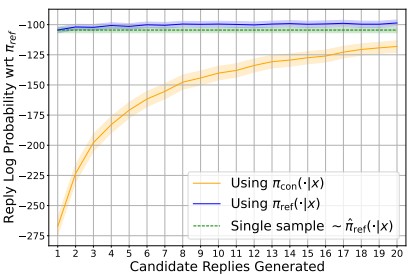

(c) Log probability w.r.t. $\pi_{\text{ref}}$ for 5 advertisers.

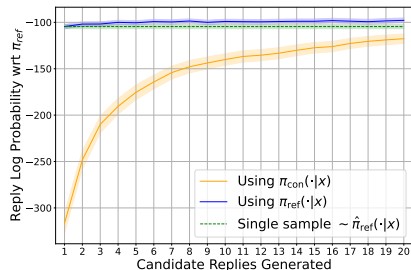

(d) Log probability w.r.t. $\pi_{\text{ref}}$ for 10 advertisers.

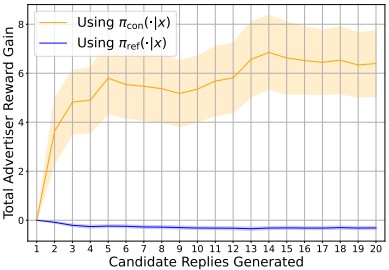

(e) Advertiser value gain for 5 advertisers.

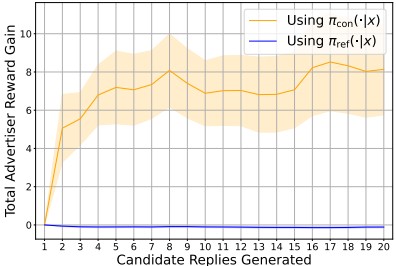

(f) Advertiser value gain for 10 advertisers.

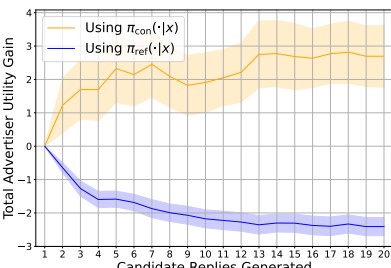

(g) Advertiser utility gain for 5 advertisers.

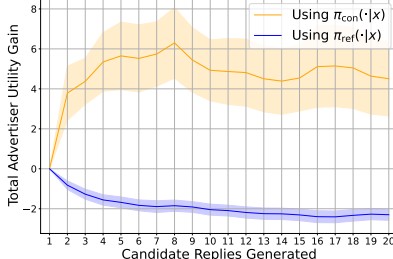

(h) Advertiser utility gain for 10 advertisers.

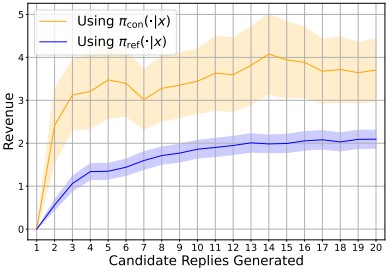

(i) Revenue for 5 advertisers.

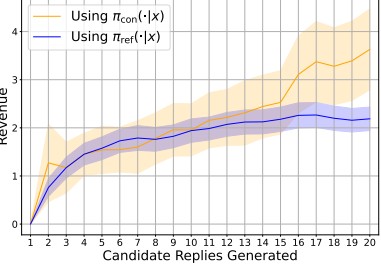

(j) Revenue for 10 advertisers.

Figure 8: Comparison of results for 5 advertisers (left) and 10 advertisers (right). Each row corresponds to a specific metric: log probability with respect to the optimal LLM $\pi_r^*$, the reference LLM $\pi_{\text{ref}}$, advertiser value and utility gain, and revenue. Shown are averages over 1000 instances including 95% CIs.

## D.7 Comprehensive Experimental Evaluation of Context

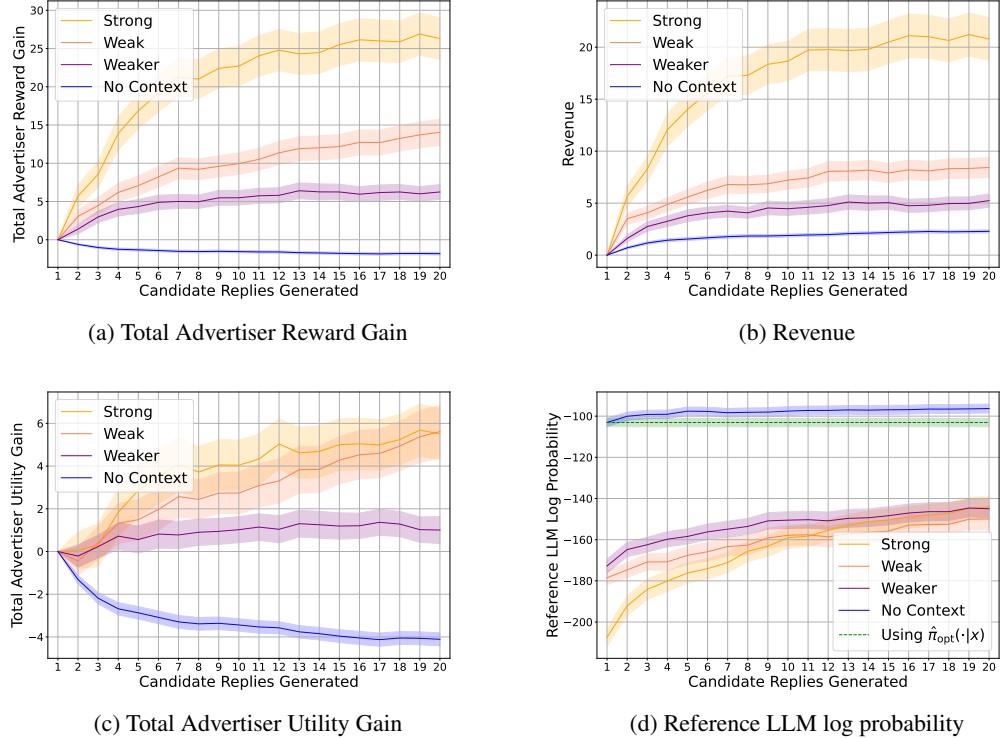

(a) Total Advertiser Reward Gain

(b) Revenue

(c) Total Advertiser Utility Gain

(d) Reference LLM log probability

Figure 9: Total advertiser normalized reward, revenue, total advertiser utility gain and reference LLM log probability as functions of the number of candidate replies generated for various levels of context. Averages over 700 runs with 95% CIs.

In this section, we experimentally evaluate MOSAIC's performance under varying levels of *context c*. Specifically, we test four levels of context:

- *Strong*: the context-aware LLM $\pi_{\text{con}}(\cdot \mid x; c)$ has access to both the advertisers' names and their descriptions;
- *Weak*: $\pi_{\text{con}}(\cdot \mid x; c)$ has access only to the advertisers' names;
- *Weaker*: $\pi_{\text{con}}(\cdot \mid x; c)$ is given neither names nor descriptions, and is simply instructed to "try to promote related brands";
- *No context*: corresponds to omitting context entirely.

Using "strong" context corresponds to the context-aware version of MOSAIC evaluated in Section 6, while "no context" corresponds to the baseline version.

For each level of context, we replicate the experiments from Section 6 using the same set of test instances. To reduce computational costs, we evaluate only 700 instances for the *strong*, *weak*, and *weaker* context levels. For the *no context* setting, we reuse the results from the 1250 instances reported in Section 6.

In Figure 9, we plot, for each context level, the total advertiser normalized reward and utility gain from participation, log probability of the reply under the optimal LLM (Equation (2)), and the platform's revenue, all as functions of the number of candidate replies generated.

In Figure 9a, we observe that for all context levels except "no context," the total advertiser reward increases rapidly with the number of generated replies. Comparing these results with the total advertiser utility gain in Figure 9c reveals an interesting insight: even under the weak context setting, where the context-aware LLM is provided only with advertiser brand names, MOSAIC generates substantial utility for advertisers, which also scales quickly with the number of candidate replies. Notably, the utility gains under strong and weak contexts using twenty candidate replies are nearly

identical. As the level of context increases beyond the weak setting, the additional social welfare generated by MOSAIC is largely recaptured as revenue for the platform, as shown in Figure 9b.

Finally, in Figure 9d, we observe that as the level of context increases, the reference LLM log probability is initially lower for small numbers of candidate replies but quickly matches across all context levels (except no context) as the number of candidates increases. This shows that MOSAIC's allocation rule (Algorithm 1) is powerful enough that the gains in advertiser value and platform revenue enabled by richer context do not come at the expense of user utility.

To conclude, MOSAIC delivers substantial advertiser utility and platform revenue even with minimal contextual information. As the level of context increases, both advertiser value and platform revenue grow significantly, without compromising the usefulness of responses to the user.

### D.8 Compute Experiments

In this section, we experimentally evaluate MOSAIC's computational requirements. Using the setup described in Section 6.1, we measure the total wall time required by MOSAIC on a *single* A100 GPU, focusing on the number of candidate replies generated for evaluation. This total time includes:

1. Generating candidate replies using the context-aware LLM.
2. Evaluating these replies with the advertisers' LLMs implemented as discussed in Section 6.1.
3. Calculating the Rochet payments, as detailed in Section 5. In practice the time to calculate payments is negligible (under 10 milliseconds), and need not increase user-perceived latency because they can be calculated after showing outputs to the user.

Figure 10 presents the total time taken by MOSAIC, which includes both the generation and evaluation of candidate replies. These components are further broken down in Figures 11 and 12, respectively.

As previously demonstrated in Section 6 and Appendix D.6, MOSAIC converges to the optimal distribution using only 20 candidate replies, regardless of the number of advertisers. In Figure 11, we observe that generating 10 and 20 candidate replies on a single A100 GPU takes 30.5 and 60.5 seconds, respectively. This represents a 2.5- and 5-fold increase over the 12.0 seconds required to generate a single reply from the same LLM, i.e., the latency perceived by the user when directly querying the LLM.

To reduce user-perceived latency, MOSAIC can parallelize candidate generation across multiple GPUs. For example, distributing the generation of 20 replies across two GPUs (each generating 10 replies) reduces the total time to 30.5 seconds, assuming efficient evaluation techniques as outlined in Section 4.2. This approach results in MOSAIC requiring approximately 5 times the compute time of generating a single reply to converge, but with a perceived latency to the user of only 2.5 times that of directly querying an LLM (of the same architecture and similar size). Although our experiments were run on a single GPU, note that MOSAIC can always be parallelized further so that the user-perceived latency is the same as directly querying a *single* LLM, as explained in Section 4.2.

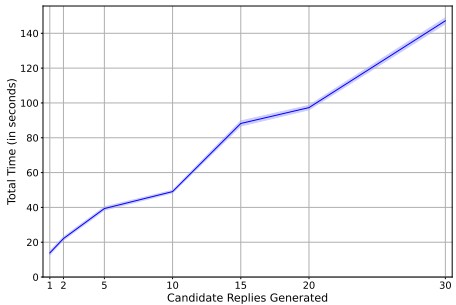

Figure 10: MOSAIC total wall time (seconds) as a function of the number of candidate replies generated using $\pi_{\text{con}}$. Shown are averages over 50 instances including 95% CIs.

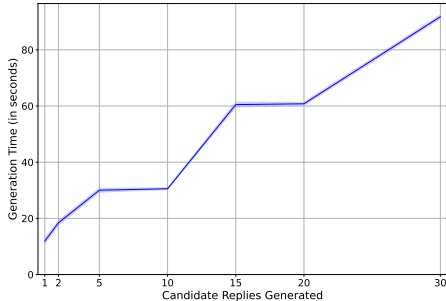

Figure 11: MOSAIC generation time as a function of the number of candidate replies generated using $\pi_{\mathrm{con}}$. Shown are averages over $50$ instances including $95\%$ CIs.

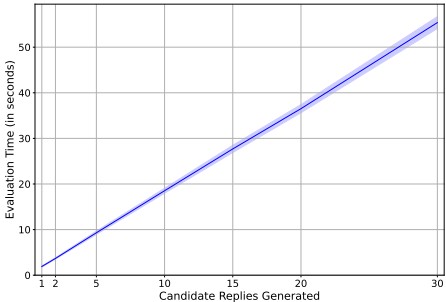

Figure 12: MOSAIC evaluation time as a function of the number of candidate replies. Shown are averages over $50$ instances including $95\ \%$ CIs.

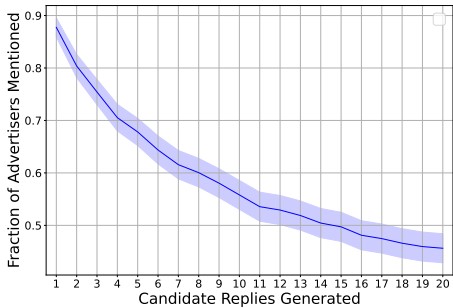

Figure 13: Fraction of advertisers mentioned in the output reply as a function of the number of candidate replies. Shown are averages over 1250 runs including 95% CIs.

## D.9 Conflicts between advertisers

The output of MOSAIC is a single text response to the user. In trying to reflect the interests of multiple advertisers, there is a risk of a single response being incoherent, just mentioning as many advertisers as possible without being a good promotion for any of them. However, the platform's objective, which takes into account both advertiser utility and usefulness for the user, should account for this: if advertisers are happy to be cross-promoted, those outcomes should be more likely, but incoherent advertisements or those that simultaneously promote competitors should be less likely.

In Figure 13 we show the fraction of advertisers mentioned by the context-aware version of MOSAIC as a function of the candidate number of replies used. We observe that initially, the fraction of advertisers mentioned is very high, but drops significantly once the mechanism has converged. The reason is precisely the fact that mentioning more advertisers in this case would cause either a significant drop in advertiser utility, or a drop in the usefulness of the reply to the user. By placing more weight towards the advertisers (i.e., decreasing the value of $\tau$ in Equation (1)), the platform can increase the number of advertisers shown when the mechanism has converged.

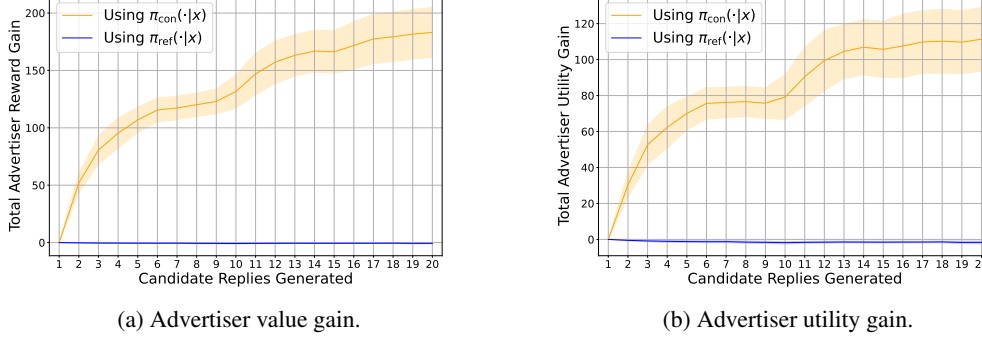

(a) Advertiser value gain.  (b) Advertiser utility gain.

Figure 14: Advertiser gains as a function of the number of candidate replies generated using $\pi_{\text{ref}}$ and $\pi_{\text{con}}$ for the flan-t5-large model. Shown are averages over 1250 instances including 95% CIs.

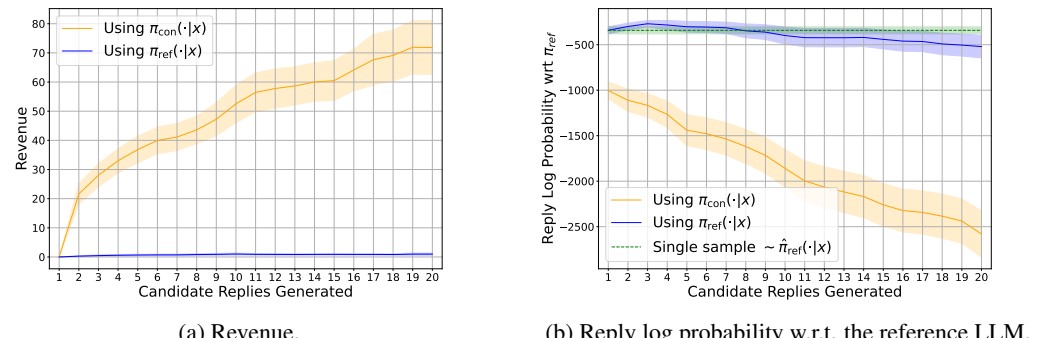

(a) Revenue.  (b) Reply log probability w.r.t. the reference LLM.

Figure 15: Revenue and reply log probability as a function of the number of candidate replies generated using $\pi_{\text{ref}}$ and $\pi_{\text{con}}$ for the flan-t5-large model. Shown are averages over 1250 instances including 95% CIs.

### D.10 Testing with additional language models

In this section, we test MOSAIC's performance using Google's flan-t5-large model [Chung et al., 2022], which uses the Apache 2.0 License.

We repeat exactly the same procedure as detailed in Section 6.1, but using flan-t5-large as the basis for both $\pi_{\text{gen}}$, the advertisers' LLMs, and the reference LLM. We use the same 50 user queries, each with two advertisers, and test each query on 10 different random seeds, resulting in 500 instances.

In Figure 14a to Figure 15b, we present the total advertiser value gain, reward gain from participating in MOSAIC, platform revenue, and the usefulness of the returned reply to users as functions of the number of candidate replies generated.

Consistent with the main paper's results using Llama-2-7b-chat-hf [Touvron et al., 2023], the context-aware version of MOSAIC demonstrates significant value and utility generation for advertisers and successfully captures a substantial portion as platform revenue. In contrast, though, the naive version of MOSAIC fails to generate value or utility for advertisers, resulting in negligible revenue for the platform. Similar to the results of the main paper, despite both mechanisms converging in the limit to the platform's objective, as shown in Corollary 4.1, their performance for a computationally tractable number of generated replies diverges.

Similar to the findings in the main paper, the naive version retains the reference LLM's usefulness to the user (which is optimal for the task), while the context-aware version prioritizes the platform's objective, leading to lower user usefulness. Overall, our results using flan-t5-large agree closely with those obtained using Llama-2-7b-chat-hf, demonstrating consistency across models. The main difference is that the context-aware mechanism is even more important when using flan-t5-large.

# E  Expanded text of prompts

Table 3: Table of user prompts with corresponding advertisers and their descriptions. Only the first two advertisers are shown for each query; the full prompts are available in the code supplement.

**User Query 1**
Learning to play the guitar.
**Advertisers**
- StrumMaster: selling guitars, tuners, and learning materials for beginners
- Chordify: offering online guitar lessons and interactive practice tools

**User Query 2**
Best skin care routine for dry skin.
**Advertisers**
- HydraSkin: providing moisturizers and serums for dry skin
- GlowEssentials: selling natural and organic skin care products

**User Query 3**
Improving productivity while working from home.
**Advertisers**
- SpaceMakers: offering ergonomic furniture and home office setups
- FocusApp: providing productivity apps and time management tools

**User Query 4**
Starting a small online business.
**Advertisers**
- EcomLaunch: offering e-commerce platform solutions and web design services
- MarketMover: providing digital marketing services and SEO optimization

**User Query 5**
Healthy meal planning on a budget.
**Advertisers**
- BudgetBites: selling affordable meal kits and recipe books
- NutriSaver: offering discounts on healthy groceries and food delivery services

**User Query 6**
Mastering digital photography.
**Advertisers**
- PixelPro: selling cameras, lenses, and photography accessories
- EditCraft: offering photo editing software and online tutorials

**User Query 7**
Effective ways to reduce household energy use.
**Advertisers**
- EcoSave: offering energy-efficient home appliances and lighting solutions
- InsulaTech: providing home insulation and energy audit services

**User Query 8**
Finding the perfect hiking trails.
**Advertisers**
- TrailFinder: offering a mobile app with detailed maps and trail reviews
- GearUp: selling outdoor gear and apparel for hiking enthusiasts

**User Query 9**
Building a personal brand on social media.
**Advertisers**
- BrandBuilder: offering personal branding courses and social media strategy consultations
- VisualizeMe: providing graphic design services for social media content

**User Query 10**
Learning a new language effectively.
**Advertisers**
- LingoLeap: offering online language learning courses and tutoring

- SpeakEasy: providing language learning apps with speech recognition technology

**User Query 11**
Staying fit without a gym.
**Advertisers**
- HomeFit: selling home workout equipment and fitness accessories
- MoveIt: offering online fitness classes and personal training sessions

**User Query 12**
Eco-friendly travel options.
**Advertisers**
- GreenPath: offering eco-friendly travel packages and sustainable tourism experiences
- EcoStay: providing listings for green hotels and accommodations

**User Query 13**
Mastering the art of cooking steak.
**Advertisers**
- GrillMaster: selling premium grills and barbecue accessories
- SteakPerfection: offering online cooking classes focused on meat preparation

**User Query 14**
Creating a successful YouTube channel.
**Advertisers**
- VidGrowth: offering video production courses and YouTube growth strategies
- ChannelDesign: providing custom YouTube channel art and video thumbnails

**User Query 15**
Decorating your home on a budget.
**Advertisers**
- DecorDeals: selling affordable home decor and furniture
- StyleSavvy: offering interior design consultations and budget-friendly decorating tips

**User Query 16**
Managing stress and anxiety.
**Advertisers**
- CalmSpace: offering mindfulness apps and stress reduction tools
- WellnessWave: providing online therapy sessions and mental health resources

**User Query 17**
How can I make my home more energy efficient?
**Advertisers**
- EcoWise: offering smart home energy solutions and solar panels
- GreenInsulate: providing eco-friendly insulation materials and installation services

**User Query 18**
What are the best strategies for investing in the stock market?
**Advertisers**
- InvestSmart: offering investment advice and financial planning services
- StockGenius: providing an AI-powered stock market analysis tool

**User Query 19**
How can I improve my mental health?
**Advertisers**
- MindBalance: offering mental health apps and mindfulness courses
- SerenityNow: providing online counseling and therapy sessions

**User Query 20**
Where can I find organic and sustainable clothing?
**Advertisers**
- EcoWear: selling sustainable and organic clothing options
- GreenThreads: offering eco-friendly fabrics and fair-trade fashion

**User Query 21**
How do I start a vegetable garden in my backyard?
**Advertisers**

- GardenGuru: selling gardening tools, seeds, and starter kits
- PlantPerfect: offering online gardening courses and personalized garden planning

**User Query 22**

What are the best tools for remote team collaboration?

**Advertisers**

- TeamSync: offering collaboration software and project management tools
- ConnectSuite: providing video conferencing solutions and team communication platforms

**User Query 23**

How can I learn to cook healthy meals quickly?

**Advertisers**

- QuickChef: offering quick and healthy cooking classes online
- NutriFast: selling kitchen gadgets and appliances for healthy cooking

**User Query 24**

What are the best ways to save for retirement?

**Advertisers**

- FutureFund: offering retirement planning and investment services
- PensionPlus: providing personalized financial advice for retirement savings

**User Query 25**

How can I decorate my apartment on a minimalist budget?

**Advertisers**

- MinimalSpace: selling minimalist furniture and decor
- SimplicityDesigns: offering interior design consultation for minimalist living spaces

**User Query 26**

What are the best practices for data security in small businesses?

**Advertisers**

- SecureIT: offering cybersecurity solutions for small businesses
- DataGuard: providing data protection services and training

**User Query 27**

How can I reduce my carbon footprint with daily habits?

**Advertisers**

- EcoHabits: offering products and tips for sustainable living
- GreenStep: providing carbon offset subscriptions and eco-friendly lifestyle goods

**User Query 28**

What are effective ways to learn a musical instrument online?

**Advertisers**

- MusicMastery: offering online music lessons and virtual tutoring
- InstaTune: selling musical instruments and learning aids

**User Query 29**

How can I keep my pets healthy and happy?

**Advertisers**

- PetWell: offering pet health insurance and wellness products
- HappyPaws: selling organic pet food and eco-friendly toys

**User Query 30**

What's the best way to start learning coding from scratch?

**Advertisers**

- CodeBegin: offering coding bootcamps and online tutorials for beginners
- DevPath: providing interactive coding courses and learning platforms

**User Query 31**

How to find affordable eco-friendly travel destinations?

**Advertisers**

- GreenTraveler: offering eco-friendly travel guides and affordable green vacation packages
- EcoTripAdvisor: providing reviews and bookings for sustainable travel options

**User Query 32**

What are the best ways to improve indoor air quality?

**Advertisers**
- AirPure: selling air purifiers and natural air freshening solutions
- BreatheEasy: offering indoor plant subscriptions and ventilation solutions

**User Query 33**
How to make my own natural beauty products?
**Advertisers**
- NatureGlow: offering DIY beauty product kits and organic ingredients
- EcoBeauty: selling guides and supplies for creating homemade natural cosmetics

**User Query 34**
What are the best techniques for effective online teaching?
**Advertisers**
- EduTech: offering online teaching tools and educational technology solutions
- TeachMaster: providing courses on online pedagogy and digital classroom management

**User Query 35**
How to create a successful podcast from home?
**Advertisers**
- PodCastify: offering podcast creation courses and audio equipment
- SoundWave: selling podcast editing software and hosting services

**User Query 36**
What are the most effective ways to market a small business locally?
**Advertisers**
- LocalBoost: offering local SEO services and marketing strategies for small businesses
- MarketTown: providing targeted advertising solutions and local business networking opportunities

**User Query 37**
How can I optimize my home office for productivity?
**Advertisers**
- WorkZen: offering ergonomic office furniture and productivity tools
- SpaceCraft: providing home office design services and organization solutions

**User Query 38**
What are the best platforms for freelance writers to find work?
**Advertisers**
- WriteFreely: offering a job board and resources for freelance writers
- ContentConnect: providing a platform to connect writers with businesses

**User Query 39**
How to get started with meditation and mindfulness?
**Advertisers**
- MindfulStart: offering guided meditation apps and mindfulness training
- ZenSpace: selling meditation cushions and mindfulness journals

**User Query 40**
Where can I find unique handmade gifts?
**Advertisers**
- CraftCorner: selling handmade gifts and artisan crafts
- MakerMarket: offering a marketplace for unique handmade items from local artists

**User Query 41**
How can I improve my credit score quickly?
**Advertisers**
- CreditBoost: offering credit monitoring services and financial advice
- ScoreSurge: providing tools and tips for improving your credit score

**User Query 42**
What are the best apps for tracking fitness and nutrition?
**Advertisers**
- FitTrack: offering comprehensive fitness and nutrition tracking apps
- NutriSync: selling personalized nutrition plans and diet tracking tools

**User Query 43**

How to plan an environmentally friendly wedding?
**Advertisers**
- GreenBride: offering eco-friendly wedding planning services and supplies
- EcoWed: providing sustainable wedding attire and decor

**User Query 44**
What are the safest ways to travel during a pandemic?
**Advertisers**
- SafeJourney: offering travel safety kits and pandemic travel advice
- HealthTravel: providing information on safe destinations and travel insurance

**User Query 45**
How can I learn DIY home repairs and improvements?
**Advertisers**
- FixItYourself: offering online courses and tutorials for home repair
- DIYHome: selling DIY home improvement tools and kits

**User Query 46**
What are the best resources for starting a plant-based diet?
**Advertisers**
- PlantEats: offering guides and meal plans for starting a plant-based diet
- VeggieVibe: selling plant-based cookbooks and kitchen gadgets

**User Query 47**
Tips for running a marathon.
**Advertisers**
- RunFastGear: specializing in high-performance running shoes
- HydraFuel: offering electrolyte drinks and energy bars for athletes

**User Query 48**
Best practices for organic gardening.
**Advertisers**
- GreenThumb Solutions: providing organic fertilizers and pest control products
- EcoGrow: selling heirloom seeds and sustainable gardening tools

**User Query 49**
How to improve home WiFi?
**Advertisers**
- SignalBoost: which offers advanced WiFi routers and extenders
- NetWizard: providing network optimization services and support

**User Query 50**
Ways to save on travel.
**Advertisers**
- BudgetJourneys: specializing in affordable travel packages and deals
- StayLocal: offering discounts on boutique hotels and unique accommodations

