# OpenReview forum: "Truthful Aggregation of LLMs with an Application to Online Advertising"
_NeurIPS.cc/2025/Conference — NeurIPS 2025 poster_

### Official Review · Reviewer_g2Ts · 2025-06-13

**Clarity:** 3
**Significance:** 3
**Originality:** 3
**Rating:** 5
**Confidence:** 3

**Summary:**

This paper study the scenario where LLM are applied to ad auctions and they propose a mechanism that converges to optimal distribution. They conduct experiments of MOSAIC in real settings. Their payment rule ensures that there is no incentive for a participant to misreport their preference, according to the theory analysis.

**Questions:**

How in real world that this model makes sense, i.e., the auctioneers pay to influence the output of LLM.

**Ethical Concerns:**

["NO or VERY MINOR ethics concerns only"]

**Final Justification:**

I have read the authors’ response defending their model. Though it is a solid empirical work, the conclusion appears to be that there is currently no mature application scenario for paying to influence an LLM’s output. I maintain my original score of acceptance.

**Limitations:**

yes

**Paper Formatting Concerns:**

/

**Quality:**

3

**Strengths And Weaknesses:**

Strength:

(1) Applying LLM to ad auctions is an emerging field and has own its significance in real application and requires further research. This paper provides one way of doing so.

(2) The results provided in this paper are comprehensive, in the sense of with both theory analysis and empirical results.

---

> ### Author Rebuttal · Authors · 2025-07-28
>
> We thank the reviewer for their positive comments about the significance of our application and thoroughness of the analysis.
>
> We assume that the main question is whether or not allowing *advertisers* to pay to steer LLM responses could have any real-world application. Currently, many LLM platforms are experimenting with advertising, although as far as we know, none allow advertisers to bid on generated content, yet. Also, one of the leading AI companies is also a leading ad auction company (Google). And in other settings (e.g., newsfeeds), running complex auctions to mix user content with ads is already standard. So we see this as the natural next step.
>
> We think our approach makes the most sense for "LLM search summaries" (e.g. Perplexity, Google AI search, etc.) where a user makes a single fact-based query and receives a reply, and where traditional search already uses an ad-driven model.

---

> > ### Comment · Reviewer_g2Ts · 2025-08-04
> >
> > Thanks the authors for the response! I resume my ratings for this paper.

---

### Official Review · Reviewer_SNPf · 2025-06-29

**Clarity:** 3
**Significance:** 2
**Originality:** 3
**Rating:** 4
**Confidence:** 3

**Summary:**

This paper introduces MOSAIC, a novel auction mechanism designed for the truthful aggregation of multiple advertisers' preferences concerning Large Language Model (LLM) outputs. The mechanism addresses the challenge of enabling advertisers to influence LLM-generated content while simultaneously allowing the platform to optimize advertiser utility without compromising user satisfaction. MOSAIC offers several key advantages: it incentivizes truthful preference reporting, provably and efficiently converges to the optimal LLM output without requiring fine-tuning or direct manipulation of model weights, and effectively integrates contextual information to enhance performance. Experimental evaluations underscore MOSAIC's effectiveness, demonstrating strong advertiser value, robust platform revenue, and notable computational efficiency.

**Questions:**

1. How robust is the MOSAIC mechanism in scenarios where advertiser preferences significantly diverge from the user preferences encapsulated by the reference LLM?

2. What implications arise for MOSAIC's performance or theoretical guarantees if the contextual policy ($\pi_{con}$) is substantially different from the optimal policy ($\pi_{opt}$)? In the current experiments, the observed difference between $\pi_{con}$ and $\pi_{opt}$ appears relatively small.

3. Do the authors believe that the current allocation rule presented is the unique solution for this problem setting?

**Ethical Concerns:**

["NO or VERY MINOR ethics concerns only"]

**Final Justification:**

I have read the authors' rebuttal, and I would like to maintain my original assessment. Specifically, I believe this paper addresses a significant problem in online advertising, and the solution can be efficiently implemented. Meanwhile, the contribution of the theoretical results remains incremental to me. **Therefore, I stand by my initial score of weakly accept.** I respect the final decision of the AC and the perspectives of the other reviewers.

**Limitations:**

yes

**Quality:**

3

**Strengths And Weaknesses:**

### Strengths:

1. This paper investigates an interesting and highly practical problem with strong motivation, particularly relevant for online advertising scenarios.

2. The authors provide thorough explanations for each of their results and conduct detailed comparisons with prior work across various properties, significantly improving the paper's clarity and ease of comprehension.

3. The proposed mechanism is straightforward to implement, and the experimental section offers valuable insights into how the size of the candidate set ($M$) influences other critical variables.

### Weaknesses:

My primary concern lies with the relative technical strength of the paper. The main theoretical result, Theorem 5.1, appears derivable by simply validating the cyclic monotonicity of the allocation rule, which in turn directly implies the payment rule. While other theoretical results pertain solely to computational aspects.

Regarding the experimental section, the comparison between the Context-Aware Mechanism Output and Baseline Mechanism Output in Figure 1 is arguably meaningless. Without incorporating advertiser-specific information, it is unreasonable to expect the LLM to generate content relevant to them.

---

> ### Author Rebuttal · Authors · 2025-07-28
>
> We thank the reviewer for acknowledging the relevance of our problem and the thoroughness of our analysis and experiments. We also especially appreciate the remark on how straightforward our mechanism is to implement: this was a primary design goal.
>
> ## Responses to questions:
>
> ### "Robustness to cases where the reference LLM diverges significantly from the advertiser’s preferences"
>
> From a theoretical perspective, MOSAIC remains truthful in all such cases. As far as payments go, if the result strongly favors the advertisers, compared to the reference LLM, they will have to pay a lot; if the result strongly favors the user, the advertisers will not pay too much . This intuition is formalized in Lemma 5.2, where we prove that an advertiser’s expected utility is proportional to how well-aligned her preferences are with the theoretically optimal LLM for the platform’s objective in the marginal economy where she does not participate in the mechanism. Put differently, the more an advertiser “moves” the output distribution, the higher her payments, and the lower her utility.
>
> ### “What implications arise for MOSAIC's performance or theoretical guarantees if the contextual policy $\pi_{con}$ is substantially different from the optimal policy $\pi_{opt}$? In the current experiments, the observed difference between πcon and πopt  appears relatively small.”
>
> In Lemma 4.2, we prove that, under very reasonable assumptions (absolute continuity),  any contextual policy will converge to the optimal policy. At the same time, in Lemma B.3, we show that the convergence rate is at least 1 / \sqrt{M}. However, the convergence also depends on the distance between the optimal and the contextual policy, meaning that, the further apart that the two policies are,  the more samples are required.
>
> We disagree that  in the current experiments, the difference between the two distributions is small: note that the y-axis of  figure 2 (a) is in log scale. Nevertheless, in appendix D.7, we perform an extensive ablation experiment on MOSAIC’s performance for various levels of context, and show that MOSAIC maintains very strong performance even under much weaker contextual policies.
>
> However, both the advertiser’s utility, and the platform’s revenue are monotonically increasing with better contextual policies. Thus, all participants in the system are incentive-aligned to provide useful context to the platform. In our eyes, this is similar to online advertising, where advertisers are incentivized to design better ads to increase their CTR and therefore their utility.
>
>
> ### Do the authors believe that the current allocation rule presented is the unique solution for this problem setting?
>
> When working on this paper, we explored many alternative approaches before settling on this design. One decision we made is that our mechanism should be static: all “options” for the advertisers are decided in advance, the advertisers report their preferences, and a final decision is taken by the mechanism. This is in contrast to a dynamic mechanism, where there are intermediate points for the advertisers to report their preferences, and then this affects the mechanism’s downstream actions. In section 2, we explain some limitations behind the dynamic mechanism for this problem proposed  by Dütting et al., 2024 [1]. To a large extent, these issues are inherent to any dynamic mechanism and not unique to that paper.
>
> We also decided that, given the advertising application we envision, any fine-tuning of weights would be impractical and computationally costly. Given this constraint, in order to change the output distribution, one has to choose some LLM sampling scheme.
>
> Considering only static and sampling-based mechanisms leaves us in a relatively small part of the design space. An approach based on importance sampling is the most natural and straightforward way to solve the problem while ensuring cyclic monotonicity. It is also pleasantly simple and works well in practice.
>
> With that being said, this does not exclude the design of sampling rules tailored to this problem.  There are many methods for guided LLM sampling appearing all the time. Some may be well-suited to conversion into an even better truthful static mechanism, in the same spirit as our paper. And there may be ways to overcome the challenges of designing dynamic incentive compatible mechanisms. This is an interesting area of future work. We will add a discussion to this end to the final version of the paper.
>
>
> ### "My primary concern lies with the relative technical strength of the paper. The main theoretical result, Theorem 5.1, appears derivable by simply validating the cyclic monotonicity of the allocation rule, which in turn directly implies the payment rule."
>
> We believe that our answer above also sheds light on your concern regarding technical strength. Cyclic monotonicity is the necessary and sufficient condition for strategyproofness, and (up to a constant) pins down truthful payments, so showing that any mechanism is strategyproof will involve nothing more or less than establishing cyclic monotonicity.
>  We see our contribution not as a highly technical proof of this property, but as designing a very simple and practical mechanism that has many nice properties including strategyproofness. While some of the other allocation rules we explored might have resulted in a more interesting proof, they were also much worse in terms of computational efficiency and practicality.
>
> With that being said, as we alluded to before, Rochet’s result identifies our payments only up to a constant. In Appendices C.3 and C.4,  we detail how naively setting that constant to zero would be detrimental to the long-term success of a mechanism for this setting, as it could lead to free-riding, while at the same time, the mechanism would not be individually rational. Then, in Section 5.3, we design an advertiser-specific payment offset that, while maintaining strategy-proofness, completely alleviates those issues. In Lemmata C.1 and C.2, we prove that MOSAIC is “almost IR”, which we explain  is the strongest notion of IR one could hope for in this setting. Additionally, in Lemma 5.2, we prove that this offset aligns each advertiser’s utility with her contribution to social welfare. Finally, in Section 6 and Appendix D, we experimentally  show how effective this offset is in mitigating these potential issues. We believe that identifying and mitigating these issues is a further important technical contribution.
>
> ### Meaningfulness of figure 1:
>
> The goal of Figure 1 was to clearly establish  that the context is important for reasonable practical performance (we also emphasize this in the experiments). This fact is the reason we need to design a mechanism that still works with samples from any \pi_con, not just \pi_ref, which is a key aspect of our paper.
>
>
> ### References
> [1] Paul Dütting, Vahab Mirrokni, Renato Paes Leme, Haifeng Xu, and Song Zuo. Mechanism Design for Large Language Models. In Proceedings of the ACM Web Conference 2024 (WWW ’24), pp. 144–155. ACM, 2024.

---

> > ### Comment · Reviewer_SNPf · 2025-08-01
> >
> > I appreciate the authors' detailed responses to my comments. After reviewing the clarifications, I maintain my original assessment of *borderline accept* and am pleased to see this paper accepted.

---

### Official Review · Reviewer_riQC · 2025-06-30

**Clarity:** 3
**Significance:** 4
**Originality:** 3
**Rating:** 4
**Confidence:** 4

**Summary:**

This paper studies a model where multiple LLMs need to truthfully aggregate preferences, with respect to a reference LLM. The authors noted that online advertising is the most natural application setting of this model, so discussion of this paper is centered around advertising.

Under the authors' model, each advertiser has her own LLM, and the advertisers aim to influence the reference LLM (provided by the search engine to give good answers to users maximizing user experience). The authors noted a relationship between the studied model and Reinforcement Learning from Human Feedback. The core technique is described in Equation 2, which basically uses RLHF techniques. To summarize it in English, we can use the refernece LLM to generate samples, and then apply importance sampling techniques to create samples that maximize Equation 1, which considers the advertisers' preference.

**Questions:**

Could you please comment on whether free riding is an issue, as a result of providing context using the method in footnote 4.

**Ethical Concerns:**

["NO or VERY MINOR ethics concerns only"]

**Final Justification:**

The authors' rebuttal addressed my concern on free-riding.

**Limitations:**

Yes, in the paper checklist, the authors emphasized that this is theoretical research.

**Paper Formatting Concerns:**

No formatting concerns

**Quality:**

3

**Strengths And Weaknesses:**

Overall, the paper's proposal of the new LLM advertising framework is, for the most part, practically sound. The algorithm is elegant. The authors showed that the allocation rule is monotonic so strategy-proof payments can be derived and different versions of IR conditions are discussed in the paper. Even though LLM advertising's main goal should be revenue in my opinion, but welfare maximizing is also a reasonable objective. Section 4.2 discusses practical considerations, which in my opinion adds a lot of strength to the paper. The proposed framework is easy to implement, unlike the earlier token-based auction, which in its current form cannot be used in practise. For example, the proposed framework uses 20 LLM forward passes and the welfare is already quite good, as shown in experiments, and the token-based auction would require a lot more to generate any long paragraph.

The downside of this paper is "context-aware" part. In practise, if we need to provide context like footnote 4 -- "try to mention advertiser x, advertiser x description", then I think it creates a free-rider situation. It will be a problem if the objective is revenue maximizing. If we need to provide context like this, then I think there is a good chance that an advertiser gets free advertisement. What I mean is that an advertiser can bid 0 and there is still chance of having the ads shown if the context is provided this way in practise. This is difficult to fix. As the authors noted, we kind of must provide the context in practise, because the beautiful theoretical result in Equation 1 and 2 cannot be used directly without the context.

---

> ### Author Rebuttal · Authors · 2025-07-28
>
> We appreciate the reviewer’s positive comments about the elegance and practicality of our algorithm. Designing a mechanism that is practical and simple to implement, yet has nice theoretical properties, was a central goal in working on this paper.
>
> ## Response to questions:
>
> ### "I think there is a potential issue of free-riding."
>
> Thank you for pointing to the potential issue of free-riding, which was indeed a major challenge we had to address.
> In Section 5.2 and Appendix C.2, we explain in detail why free-riding can be much more pronounced in LLM-based advertising compared to traditional online advertising. In Appendix C.4, we additionally explain why free-riding will lead to market unravelling for any mechanism in this setting.
>
> To mitigate this potential issue, in Section 5.3, we introduce a carefully-designed offset to the Rochet payments, ensuring that MOSAIC remains truthful, while satisfying a property we call "what you give is what you get:"with this offset in place, advertisers' utilities are proportional to their contributions to MOSAIC's welfare. Thus, advertisers who have very low values also have very low utility, which effectively mitigates the concern regarding “free-riding.”
>
> Finally, in Appendix D.5, we provide strong computational experiments, showing that, in all cases tested, this advertiser-specific offset increases an advertiser's expected utility, makes the relationship between an advertiser's contribution to social welfare and her utility both more linear and positively correlated, and also makes  our mechanism ex-ante IR.

---

> > ### Comment · Reviewer_riQC · 2025-08-05
> >
> > Thank you for the clarifications. I will maintain my original score.

---

> > ### Comment · Reviewer_vqqH · 2025-08-09
> >
> > Thanks for the detailed response, which have resolved some of my concern. I have carefully read the response from the authors and the comments from other reviewers. I would still keep my score. Thanks.

---

### Official Review · Reviewer_vqqH · 2025-07-13

**Clarity:** 3
**Significance:** 3
**Originality:** 3
**Rating:** 5
**Confidence:** 4

**Summary:**

This paper introduces a truthful mechanism MOSAIC for advertising with LLMs. The authors assume that each advertiser provides an LLM or reward model as the ad creation and wants to affect the auctioneer’s LLM output towards the advertiser’s benefit. This paper provides a practical mechanism to balance the advertisers’ utilities and user’s satisfaction. The mechanism includes an allocation rule that only requires API access to the advertisers’ models and approximately maximizes the social welfare, and a payment rule that incentivizes truthful reward reporting. The authors conducted experiments on real LLMs, demonstrating the effectiveness of MOSAIC.

**Questions:**

When the advertiser evaluates its preference of the query-answer pair, the user query is meant to be sent to the advertiser’s model. This can potentially cause privacy issues, especially in the “API access” scenario.
How about the cost of serving the advertisers’ models? The cost could be too high to be ignored in the mechanism especially for the cutting-edge LLMs.

**Ethical Concerns:**

["NO or VERY MINOR ethics concerns only"]

**Final Justification:**

I have read the rebuttal and discussions with authors, and maintain my current rating.

**Limitations:**

please refer above

**Quality:**

3

**Strengths And Weaknesses:**

Strength
To the best of my knowledge, this work proposed one of the first practical mechanisms for LLM advertising, showing the potential to fill the gap between theory and practice. The mechanism is concise and theoretical analysis is solid.
Weaknesses
Considering that the authors claim this is a practical mechanism, the experiment setup is not sufficient, leading to less convincing results. For example, the authors did not discuss the optimal M choice when various numbers of advertisers participated in. I suggest the authors provide more experimental results in various settings, including different kinds of advertiser’s model (reward model or LLMs), various ad topics and more advertisers.
The paper assumes advertisers can accurately encode their preferences via LLMs or reward functions, which may not be feasible for all advertisers, especially smaller or non-technical ones.
Some of the notations are not clear. For example, what is the exact definition of $\pi_{\hat{r}_{-i}}$ in Lemma 5.2 (Line 287)?.
The paper does not deeply engage with ethical implications of allowing advertisers to influence LLM outputs, such as manipulation, bias amplification, or impacts on user trust. There is also little discussion of fairness, such as whether small advertisers get fair exposure compared to large ones with higher rewards.

---

> ### Author Rebuttal · Authors · 2025-07-28
>
> We thank the reviewer for their positive comments about the relevance of our application and their positive remarks about our mechanism. We worked hard to make a mechanism without needless complexity, yet which works well and has the right theoretical properties.
>
> ## Response to questions:
>
> ### “The authors did not discuss the optimal M choice when various numbers of advertisers participated”
>
> In appendix D.6 we provide results with a varying number of advertisers and show that, for all numbers of advertisers tested, MOSAIC's performance across all metrics of interest increases monotonically with M. At the same time, the marginal benefit of increasing M is decreasing, as supported by our theoretical analysis. The optimal M  to use would be the one where the marginal cost of generating an additional reply becomes greater than the marginal benefit to the platform  (see appendix D.8 for a detailed analysis of the marginal cost of increasing M). Of course, that depends on the business weights placed by the platform on the various metrics of interest.
> Note that this trade-off is not unique to advertising with LLMs; in classic online ad auctions, large platforms have to trade off the predictive power of the models they use to evaluate the quality of ads shown to the user with the computational costs of those same models.
>
>
> ### Further experimental results (different reward models, various numbers of advertisers,  different ad topics):
>
> We provide extensive additional experimental results throughout the appendix:
>
> Reward models: In Appendix D.10, we evaluate multiple LLM architectures (corresponding to different reward models) and observe consistently strong performance. That said, identifying which reward models best reflect advertiser preferences remains an important direction for future work.
>
> Number of advertisers: In Appendix D.6, we provide extensive experimental results varying the number of advertisers, and observe that MOSAIC exhibits similarly strong performance for all tested setups.
>
> Query diversity: Our experiments use over 50 user queries spanning diverse topics and advertiser types including music lessons, skincare routines, household utility providers, and mental health advice. For a complete list of prompts and participating advertisers please see Appendix E.
>
>
>
> ### On the feasibility of  advertisers accurately encoding their preferences via LLMs and on potential privacy issues on the “API access” scenario:
>
> Thank you for raising these very reasonable concerns. To address the first point: An advertiser indeed has higher utility in our mechanism if they can produce LLMs that better reflect their interests. This is generally the case in truthful mechanism design: a participant’s utility is maximized when they perfectly report their value function, which is why for our theoretical analysis we adopt the typical mechanism design assumption that truthful reporting is possible.
>
> Of course, from a practical perspective, a real advertiser might not have the know-how to produce an LLM reflecting their advertising preferences. This is closely related to two interlocking issues in conventional online ad auctions: (1) the conversions that an advertiser will experience also depend on the quality of the ad, not just the position where it is shown, and (2) advertisers may not know how to bid appropriately in the auction. Ad platforms offer various solutions to help advertisers improve their ads, and autobidders to help them bid successfully, an approach that could also be adopted for our mechanism.
>
> Several of the largest online advertising platforms also have the capacity to host and finetune LLMs, so they already have the infrastructure to offer an “LLM autobidder”, where they would help an advertiser train an LLM to reflect their interests, and also provide inference. Offering such a feature would be consistent with longstanding practice in traditional ad auctions. And, if all LLMs are hosted by the platform, it would solve the issue of user privacy.
>
> We note that the platform-hosted setup described here is simply an alternate framing of the same mechanism. All theoretical guarantees discussed in the paper are preserved. We will add a short discussion of this alternate perspective on the mechanism to the final version of the paper.
>
>
> ### “How about the cost of serving the advertisers’ models? The cost could be too high to be ignored in the mechanism especially for the cutting-edge LLMs.”
>
> There is a cost in extra inference passes that would need to be offset by the revenue brought in by the auction. In Appendix D.8, we provide detailed computational results, showing that the compute cost of generating 10 or 20 candidate replies (sufficient to converge to the optimal distribution for all number of advertisers tested) is less than 2.5 and 5 times the compute cost of generating a single reply. At the same time, in Section 4.2, we detail how MOSAIC is fully compatible with efficient methods for query generation and evaluation, which means that it benefits from any advancements in this research area. Finally, in practice, we don’t think that a high-parameter model that also uses a ton of reasoning tokens (for example) would be required to accurately capture advertiser preferences.
>
> It is also worth noting that online ad platforms incur per-query inference costs to serve ads. For instance, in sponsored search auctions, the platform estimates the expected value of showing a specific ad in a specific position by performing inference to predict the likelihood of a user conversion. This is conceptually similar to evaluating advertiser-specific models in our setting. In that sense, MOSAIC’s inference requirements are compatible with existing infrastructure and practices.
>
>
> ### "What is the exact definition of $\pi_{\hat{r}_{-i}}$?"
>
> $\pi_{\hat{r}_{-i}}$ is the theoretically optimal LLM for  the reported reward profile of all advertisers other than $i$. We will clarify this in the final version of the paper.
>
>
> ### Alternative objectives such as fairness for smaller advertisers:
>
> The most common objective in mechanism design, and by extension in ad auctions, is social welfare. Creating a similar mechanism for a different objective, e.g. egalitarian social welfare or a similar concept, would be valuable follow-up work, and in other mechanisms design settings such mechanisms have followed welfare-maximizing solutions. But we believe that solving the mechanism design problem for social welfare is the key first step.
>
> ### On the ethical implications and impact on user trust of allowing advertisers to influence LLM outputs:
>
> We’ve made the key assumption that, in the absence of advertisers, the platform’s and user’s interests are aligned: that is, the reference LLM has been successfully trained to maximize usefulness to the user. This mirrors common assumptions in settings like sponsored search or social media feeds, where the native (non-ad) content is assumed to optimize user utility.
>
> In practice, of course, this alignment may be imperfect. To what extent do trained LLMs reflect user interests? Do users trust them? Should they? These are deep and important questions, but they are orthogonal to our paper, relating to human psychology and perception, and arise even in advertising-free settings. We view them as well outside the scope of our work.
>
> Rather than address such questions incompletely, we adopt the standard assumption in both theoretical and applied mechanism design: namely, that true user and advertiser preferences can be accurately conveyed to the mechanism.
>
> Finally, it's worth noting that these concerns are not unique to LLM-based advertising. In any platform that incorporates ads, whether search engines or social media, the presence of advertisers influences what users see. Ensuring the integrity of this ecosystem requires vetting advertiser content, a responsibility of the platform. But this vetting process is orthogonal to the question we study: how to best represent vetted advertisers within a mechanism that balances revenue for the platform, value for the advertisers, and utility for the user.
>
> We will add a brief discussion to the paper highlighting the need for more research in this direction.

---

### Note · Authors · 2025-08-14

To the AC, we only want to briefly highlight the positive consensus about our paper: there is agreement that the problem is well-motivated, and that our mechanism is simple, practical, and has desirable theoretical properties.

Thanks to all involved in the review process for your thoughtful engagement with the paper. Addressing the reviewers' questions and concerns will help make the paper stronger in several areas.

---

### Decision · Program_Chairs · 2025-09-17

**Decision:**

Accept (poster)

**Comment:**

The paper studies the problem of designing LLM-based truthful auctions, where a user queries an LLM whose answer is the result of an auction among different “advertisers” who would like to sway the outcome in their favor, whereas the platform wants to generate the best answer for the user, while ensuring high reward for the advertisers. The authors make connections with RLHF and design an allocation rule and a payment strategy to ensure a strategy proof mechanism, while maximizing social welfare. The paper reports a series of theoretical guarantees as well as a practical implementation and tests using mid-scale LLMs in a synthetic advertising scenario.

Based on the reviewers and rebuttal, there is general consensus towards accepting the paper. The main concerns raised by the reviewers relate to some of the setting and theoretical assumptions (e.g., assuming advertisers have a properly trained LLMs for their preferences) and the empirical validation (e.g., number of advertisers). The authors already addressed some of these concerns in the appendix and in the rebuttal, but there are definitely some remaining aspects that are not fully covered such as ethical considerations, a broader fairness analysis, or privacy issues. While I believe these are very valid concerns, the novelty of the contribution, its technical soundness and preliminary evidence of effectiveness outweighs these concerns and make the paper a very solid “first” step in this new very significant problem. In order to reinforce even further this aspect, I strongly encourage to further expand the conclusion section by discussing all the limitations of the current setting/approach and the most promising open research directions.